# LEAD: Min-Max Optimization from a Physical Perspective

**Reyhane Askari Hemmat***  *reyhane.askari.hemmat@umontreal.ca*
*Department of Computer Science and Operations Research*
*University of Montreal and Mila, Quebec AI Institute*

**Amartya Mitra***  *amitr003@ucr.edu*
*Department of Physics and Astronomy*
*University of California, Riverside*

**Guillaume Lajoie**  *g.lajoie@umontreal.ca*
*Department of Mathematics and Statistics*
*University of Montreal and Mila, Quebec AI Institute*
*Canada CIFAR AI chair*

**Ioannis Mitliagkas**  *ioannis@iro.umontreal.ca*
*Department of Computer Science and Operations Research*
*University of Montreal and Mila, Quebec AI Institute*
*Canada CIFAR AI chair*

**Reviewed on OpenReview:** *https://openreview.net/forum?id=vXSsTYs6ZB*

## Abstract

Adversarial formulations such as generative adversarial networks (GANs) have rekindled interest in two-player min-max games. A central obstacle in the optimization of such games is the rotational dynamics that hinder their convergence. In this paper, we show that game optimization shares dynamic properties with particle systems subject to multiple forces, and one can leverage tools from physics to improve optimization dynamics. Inspired by the physical framework, we propose LEAD, an optimizer for min-max games. Next, using Lyapunov stability theory and spectral analysis, we study LEAD's convergence properties in continuous and discrete time settings for a class of quadratic min-max games to demonstrate linear convergence to the Nash equilibrium. Finally, we empirically evaluate our method on synthetic setups and CIFAR-10 image generation to demonstrate improvements in GAN training.

## 1 Introduction

Much of the advances in traditional machine learning can be attributed to the success of gradient-based methods. Modern machine learning systems such as GANs (Goodfellow et al., 2014), multi-task learning, and multi-agent settings (Sener & Koltun, 2018) in reinforcement learning (Bu et al., 2008) require joint optimization of two or more objectives which can often be formulated as games. In these *game* settings, best practices and methods developed for single-objective optimization are observed to perform noticeably poorly (Mescheder et al., 2017; Balduzzi et al., 2018b; Gidel et al., 2019). Specifically, they exhibit rotational dynamics in parameter space about the *Nash Equilibria* (Mescheder et al., 2017), slowing down convergence. Recent work in game optimization (Wang et al., 2019; Mazumdar et al., 2019; Mescheder et al., 2017; Balduzzi et al., 2018b; Abernethy et al., 2019; Loizou et al., 2020) demonstrates that introducing additional second-order terms in the optimization algorithm helps to suppress these rotations, thereby improving convergence.

Taking inspiration from recent work in single-objective optimization that re-derives existing accelerated methods from a variational perspective (Wibisono et al., 2016; Wilson et al., 2016), in this work, we adopt

---

*Equal Contribution. Significant part of this work was done while Amartya Mitra was interning at Mila.

a similar approach in the context of games. To do so, we borrow formalism from physics by likening the gradient-based optimization of two-player (zero-sum) games to the dynamics of a system where we introduce relevant forces that helps curb these rotations. We consequently utilize the dynamics of this resultant system to propose our novel second-order optimizer for games, *LEAD*.

Next, using Lyapunov and spectral analysis, we demonstrate linear convergence of our optimizer (LEAD) in both continuous and discrete-time settings for a class of quadratic min-max games. In terms of empirical performance, LEAD achieves an FID of 10.49 on CIFAR-10 image generation, outperforming existing baselines such as BigGAN (Brock et al., 2018), which is approximately 30-times larger than our baseline ResNet architecture.

What distinguishes LEAD from other second-order optimization methods for min-max games such as Mescheder et al. (2017); Wang et al. (2019); Mazumdar et al. (2019); Schäfer & Anandkumar (2019) is its computational complexity. All these different methods involve Jacobian (or Jacobian-inverse) vector-product computation commonly implemented using a form of approximation. Thus making a majority of them intractable in real-world large scale problems. On the other hand, LEAD involves computing only *one-block* of the full Jacobian of the gradient vector-field multiplied by a vector. This makes our method significantly cheaper and comparable to several first-order methods, as we show in section 6. We summarize our contributions below:

- In section 3, we model gradient descent-ascent as a physical system. Armed with the physical model, we introduce counter-rotational forces to curb the existing rotations in the system. Next, we employ the principle of least action to determine the (continuous-time) dynamics. We then accordingly discretize these resultant dynamics to obtain our optimization scheme, Least Action Dynamics (LEAD).

- In section 4, we use Lyapunov stability theory and spectral analysis to prove a linear convergence of LEAD in continuous and discrete-time settings for quadratic min-max games.

- Finally, in section 7, we empirically demonstrate that LEAD is computationally efficient. Additionally, we demonstrate that LEAD improves the performance of GANs on different tasks such as 8-Gaussians and CIFAR-10 while comparing the performance of our method against other first and second-order methods.

- The source code for all the experiments is available at `https://github.com/lead-minmax-gam es/LEAD`. Furthermore, we provide a blog post that summarizes our work, which is also available at `https://reyhaneaskari.github.io/LEAD.html`.

## 2 Problem Setting

**Notation**   Continuous time scalar variables are in uppercase letters ($X$), discrete-time scalar variables are in lower case ($x$) and vectors are in boldface (**A**). Matrices are in blackboard bold ($\mathbb{M}$) and derivatives w.r.t. time are denoted as an over-dot ($\dot{x}$). Furthermore, off-diag$[\mathbb{M}]_{i,j}$ is equal to $\mathbb{M}_{i,j}$ for $i \neq j$, and equal to 0 for $i = j$ where $i, j = 1, 2, \ldots, n$.

**Setting**   In this work, we study the optimization problem of two-player zero-sum games,

$$\min_X \max_Y f(X, Y), \tag{1}$$

where $f : \mathbb{R}^n \times \mathbb{R}^n \to \mathbb{R}$, and is assumed to be a convex-concave function which is continuous and twice differentiable w.r.t. $X, Y \in \mathbb{R}$. It is to be noted that though in developing our framework below, $X, Y$ are assumed to be scalars, it is nevertheless found to hold for the more general case of vectorial $X$ and $Y$, as we demonstrate both analytically (Appendix C) and empirically, our theoretical analysis is found to hold.

## 3 Optimization Mechanics

In our effort to study min-max optimization from a physical perspective, we note from classical physics the following: under the influence of a net force $F$, the equation of motion of a physical object of mass $m$, is determined by Newton's $2^{\text{nd}}$ Law,

$$m\ddot{X} = F, \tag{2}$$

with the object's coordinate expressed as $X_t \equiv X$. According to the *principle of least action*[1] (Landau & Lifshitz, 1960), nature "selects" this particular trajectory over other possibilities, as a quantity called the *action* is extremized along it.

We start with a simple observation that showcases the connection between optimization algorithms and physics. Polyak's heavy-ball momentum (Polyak, 1964) is often perceived from a physical perspective as a ball moving in a "potential" well (cost function). In fact, it is straightforward to show that Polyak momentum is a discrete counterpart of a continuous-time equation of motion governed by Newton's $2^{\text{nd}}$ Law. For single-objective minimization of an objective function $f(x)$, Polyak momentum follows:

$$x_{k+1} = x_k + \beta (x_k - x_{k-1}) - \eta \nabla_x f(x_k), \tag{3}$$

where $\eta$ is the learning rate and $\beta$ is the momentum coefficient. For simplicity, setting $\beta$ to one, and moving to continuous time, one can rewrite this equation as,

$$\frac{(x_{k+\delta} - x_k) - (x_k - x_{k-\delta})}{\delta^2} = -\frac{\eta}{\delta^2}\nabla_x f(x_k), \tag{4}$$

and in the limit $\delta, \eta \to 0$, Eq.(4) then becomes ($x_k \to X(t) \equiv X$),

$$m\ddot{X} = -\nabla_X f(X). \tag{5}$$

This is equivalent to Newton's $2^{\text{nd}}$ Law of motion (Eq.(2)) of a particle of mass $m = \delta^2/\eta$, and identifying $F = -\nabla_X f(X)$ (i.e. $f(X)$ acting as a *potential* function (Landau & Lifshitz, 1960)). Thus, Polyak's heavy-ball method Eq.(3) can be interpreted as an object (ball) of mass $m$ rolling down under a potential $f(X)$ to reach the minimum while accelerating.

Armed with this observation, we perform an extension of 5 to our min-max setup,

$$\begin{aligned}
m\ddot{X} &= -\nabla_X f(X, Y), \\
m\ddot{Y} &= \nabla_Y f(X, Y),
\end{aligned} \tag{6}$$

which represents the dynamics of an object moving under a *curl force* (Berry & Shukla, 2016): $\boldsymbol{F}_{\text{curl}} = (-\nabla_X f, \nabla_Y f)$ in the 2-dimensional $X - Y$ plane, see Figure 1 (a) for a simple example where the continuous-time dynamic of a curl force (or equivalently a vortex force) diverges away from the equilibrium.

Furthermore, it is to be noted that discretization of Eq.(6) corresponds to Gradient Descent-Ascent (GDA) with momentum 1. Authors in (Gidel et al., 2019) found that this optimizer is divergent in the prototypical min-max objective, $f(X, Y) = XY$, thus indicating the need for further improvement.

To this end, we note that the failure modes of the optimizer obtained from the discretization of Eq.(6), can be attributed to: *(a)* an outward rotatory motion by our particle of mass $m$, accompanied by *(b)* an increase in its velocity over time. Following these observations, we aim to introduce suitable *counter-rotational* and *dissipative* forces to our system above, in order to tackle *(a)* and *(b)* in an attempt to achieve converging dynamics.

Specifically, as an initial consideration, we choose to add to our system, two ubiquitous forces:

- magnetic force,

$$\boldsymbol{F}_{\text{mag}} = \left(-q\nabla_{XY}f\ \dot{Y}, q\nabla_{XY}f\ \dot{X}\right) \tag{7}$$

---

[1]Also referred to as the Principle of Stationary Action.

known to produce rotational motion (in charged particles), to counteract the rotations introduced by $\boldsymbol{F}_{\text{curl}}$. Here, $q$ is the charge imparted to our particle.

- friction,

$$\boldsymbol{F}_{\text{fric}} = \left(\mu\dot{X}, \mu\dot{Y}\right) \tag{8}$$

to prevent the increase in velocity of our particle ($\mu$: coefficient of friction).

Assimilating all the above forces $\boldsymbol{F}_{\text{curl}}$, $\boldsymbol{F}_{\text{mag}}$ and $\boldsymbol{F}_{\text{fric}}$, the equations of motion (EOMs) of our crafted system then becomes,

$$\begin{aligned}
m\ddot{X} &= \boldsymbol{F}_{\text{curl}} + \boldsymbol{F}_{\text{mag}} + \boldsymbol{F}_{\text{fric}}, \\
m\ddot{Y} &= \boldsymbol{F}_{\text{curl}} + \boldsymbol{F}_{\text{mag}} + \boldsymbol{F}_{\text{fric}}.
\end{aligned} \tag{9}$$

Or equivalently,

$$\begin{aligned}
m\ddot{X} &= -\mu\dot{X} - \nabla_X f - q\nabla_{XY}f\dot{Y}, \\
m\ddot{Y} &= -\mu\dot{Y} + \nabla_Y f + q\nabla_{XY}f\dot{X}.
\end{aligned} \tag{10}$$

Without loss of generality, from hereon we set the mass of our object to be unity. In the rest of this work, we study the above EOMs in continuous and discrete time for min-max games. See Figure 1 (c) for a simple example where the continuous-time dynamic of a particle under all the above forces converges to the equilibrium.

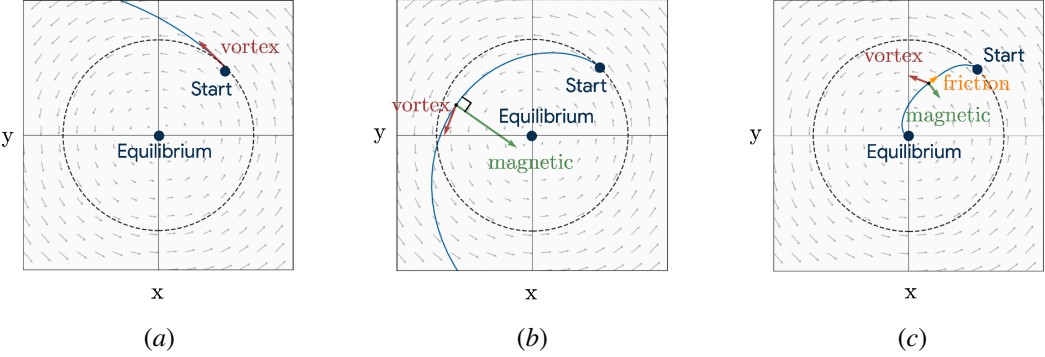

(a)          (b)          (c)

Figure 1: Depiction of the continuous-time dynamics of a particle experiencing different forces. In (a), a particle inside a vortex diverges from the equilibrium. This corresponds to the continuous-time dynamics of gradient descent-ascent with momentum in the bilinear game. In (b), we introduce a counter-rotational magnetic force to the existing system with the vortex force, assuming the particle is charged. The magnetic force is exerted in a perpendicular direction to the current direction of the particle and affects the rotations induced by the vortex. However, we do not observe convergence, which is expected from a physics perspective. The vortex force is known to increase the particle's speed over time Berry & Shukla (2016), while the magnetic force does not affect the particle's speed. Therefore, the particle's velocity will keep increasing over time, preventing convergence. To decrease the particle's velocity for convergence, we introduce friction to the system in (c). As a result, we observe that the friction causes the particle to lose speed and converge.

## 3.1 Discretization

With the continuous-time trajectory of Eq.(10) in hand, we now proceed to discretize it using a combination of Euler's implicit and explicit discretization schemes. To discretize $\dot{X} = V_X$ we have,

$$\begin{aligned}
\text{Euler Implicit Discretization}: \ & x_{k+1} - x_k = \delta v^x_{k+1} \\
\text{Euler Explicit Discretization}: \ & x_{k+1} - x_k = \delta v^x_k.
\end{aligned} \tag{11}$$

where $\delta$ is the discretization step-size and $k$ is the iteration step.

**Proposition 1.** *The continuous-time EOMs (10) can be discretized in an implicit-explicit way, to yield,*

$$
\begin{aligned}
x_{k+1} = {} & x_k + \beta(x_k - x_{k-1}) - \eta \nabla_x f\left(x_k, y_k\right) \\
& - \alpha \nabla_{xy} f\left(x_k, y_k\right)\left(y_k - y_{k-1}\right), \\
y_{k+1} = {} & y_k + \beta(y_k - y_{k-1}) + \eta \nabla_y f\left(x_k, y_k\right) \\
& + \alpha \nabla_{yx} f\left(x_k, y_k\right)\left(x_k - x_{k-1}\right),
\end{aligned}
\tag{12}
$$

where we have defined $\alpha = 2q\delta$, $\beta = 1 - \mu\delta$ and $\eta = \delta^2$ (Proof in Appendix B).

Taking inspiration from the fact that Eq. 10 corresponds to the trajectory of a charged particle under a curl, magnetic and frictional force, as governed by the principle of least action, we refer to the discrete update rules of Eq. 12 as *Least Action Dynamics (LEAD)*. (Algorithm 1 details the pseudo-code of LEAD).

**Understanding the terms in LEAD**: Analyzing our novel optimizer, we note that it consist of three types of terms, namely,

1. **Gradient Descent or Ascent**: $-\nabla_x f$ or $\nabla_y f$: Each player's immediate direction of improving their own objective.

2. **Momentum**: Standard Polyak momentum term; known to accelerate convergence in optimization and recently in smooth games. (Gidel et al., 2019; Azizian et al., 2020b; Lorraine & Duvenaud, 2022)

3. **Coupling term**:

$$
-\nabla_{xy} f\left(x_k, y_k\right)\left(y_k - y_{k-1}\right), \nabla_{yx} f\left(x_k, y_k\right)\left(x_k - x_{k-1}\right)
$$

Main new term in our method. It captures the first-order interaction between players. This cross-derivative corresponds to the counter-rotational force in our physical model; it allows our method to exert control on rotations.

---

**Algorithm 1** Least Action Dynamics (LEAD)

---

**Input**: learning rate $\eta$, momentum $\beta$, coupling coefficient $\alpha$.
**Initialize:** $x_0 \leftarrow x_{init}$, $y_0 \leftarrow y_{init}$, $t \leftarrow 0$
**while** not converged **do**
    $t \leftarrow t + 1$
    $g_x \leftarrow \nabla_x f(x_t, y_t)$
    $g_{xy}\Delta y_t \leftarrow \nabla_y(g_x)(y_t - y_{t-1})$
    $x_{t+1} \leftarrow x_t + \beta(x_t - x_{t-1}) - \eta g_x - \alpha g_{xy}\Delta y_t$
    $g_y \leftarrow \nabla_y f(x_t, y_t)$
    $g_{xy}\Delta x_t \leftarrow \nabla_x(g_y)(x_t - x_{t-1})$
    $y_{t+1} \leftarrow y_t + \beta(y_t - y_{t-1}) + \eta g_y + \alpha g_{xy}\Delta x_t$
**end while**
**return** $(x_{k+1}, y_{k+1})$

---

## 4 Convergence Analysis

We now study the behavior of LEAD on the quadratic min-max game,

$$
f\left(\mathbf{X}, \mathbf{Y}\right) = \frac{h}{2}||\mathbf{X}||^2 - \frac{h}{2}||\mathbf{Y}||^2 + \mathbf{X}^T \mathbb{A} \mathbf{Y}
\tag{13}
$$

where $\mathbf{X}, \mathbf{Y} \in \mathbb{R}^n$, $\mathbb{A} \in \mathbb{R}^n \times \mathbb{R}^n$ is a (constant) coupling matrix and $h$ is a scalar constant. Additionally, the Nash equilibrium of the above game lies at $\mathbf{X}^* = 0, \mathbf{Y}^* = 0$. Let us further define the *vector field* $\mathbf{v}$ of the above game, $f$, as,

$$
\boldsymbol{v} = \begin{bmatrix} \nabla_{\mathbf{X}} f\left(\mathbf{X}, \mathbf{Y}\right) \\ -\nabla_{\mathbf{Y}} f\left(\mathbf{X}, \mathbf{Y}\right) \end{bmatrix} = \begin{bmatrix} h\boldsymbol{X} + \mathbb{A}\mathbf{Y} \\ h\boldsymbol{Y} - \mathbb{A}^\top \mathbf{X} \end{bmatrix}.
\tag{14}
$$

### 4.1 Continuous Time Analysis

A general way to prove the stability of a dynamical system is to use a Lyapunov function (Hahn et al., 1963; Lyapunov, 1992). A scalar function $\mathcal{E}_t : \mathbb{R}^n \times \mathbb{R}^n \to \mathbb{R}$, is a Lyapunov function of a continuous-time dynamics if $\forall\, t$,

$$(i)\ \ \mathcal{E}_t(\mathbf{X}, \mathbf{Y}) \geq 0,$$
$$(ii)\ \ \dot{\mathcal{E}}_t(\mathbf{X}, \mathbf{Y}) \leq 0$$

The Lyapunov function $\mathcal{E}_t$ can be perceived as a generalization of the total energy of the system and the requirement $(ii)$ ensures that this generalized energy decreases along the trajectory of evolution, leading the system to convergence as we will show next.

For the quadratic min-max game defined in Eq.(13), Eq.(10) generalizes to,

$$\ddot{\boldsymbol{X}} = -\mu\dot{\boldsymbol{X}} - (h + \mathbb{A})\boldsymbol{Y} - q\mathbb{A}\dot{\boldsymbol{Y}}$$
$$\ddot{\boldsymbol{Y}} = -\mu\dot{\boldsymbol{Y}} - (h - \mathbb{A}^T)\boldsymbol{X} + q\mathbb{A}^T\dot{\boldsymbol{X}}, \tag{15}$$

**Theorem 1.** *For the dynamics of Eq.(15),*

$$\mathcal{E}_t = \frac{1}{2}\left(\dot{\boldsymbol{X}} + \mu\boldsymbol{X} + \mu\mathbb{A}\boldsymbol{Y}\right)^T\left(\dot{\boldsymbol{X}} + \mu\boldsymbol{X} + \mu\mathbb{A}\boldsymbol{Y}\right)$$
$$+ \frac{1}{2}\left(\dot{\boldsymbol{Y}} + \mu\boldsymbol{Y} - \mu\mathbb{A}^T\boldsymbol{X}\right)^T\left(\dot{\boldsymbol{X}} + \mu\boldsymbol{Y} - \mu\mathbb{A}^T\boldsymbol{X}\right) \tag{16}$$
$$+ \frac{1}{2}\left(\dot{\boldsymbol{X}}^T\dot{\boldsymbol{X}} + \dot{\boldsymbol{Y}}^T\dot{\boldsymbol{Y}}\right) + \boldsymbol{X}^T(h + \mathbb{A}\mathbb{A}^T)\boldsymbol{X} + \boldsymbol{y}^T(h + \mathbb{A}^T\mathbb{A})\boldsymbol{Y}$$

*is a Lyapunov function of the system.*

*Furthermore, setting $q = (2/\mu) + \mu$, we find $\dot{\mathcal{E}}_t \leq -\rho\mathcal{E}_t$ for*

$$\rho \leq \min\left\{\frac{\mu}{1 + \mu}, \ \frac{2\mu(\sigma_{\min}^2 + h)}{\left(1 + \sigma_{\min}^2 + 2h\right)\left(\mu^2 + \mu\right) + 2\sigma_{\min}^2}\right\}$$

*with $\sigma_{\min}$ being the smallest singular value of $\mathbb{A}$. This consequently ensures linear convergence of the dynamics of Eq. 15,*

$$\boxed{||\boldsymbol{X}||^2 + ||\boldsymbol{Y}||^2 \leq \frac{\mathcal{E}_0}{h + \sigma_{min}^2}\exp\left(-\rho t\right)}. \tag{17}$$

(Proof in Appendix C).

### 4.2 Discrete-Time Analysis

In this section, we next analyze the convergence behavior of LEAD, Eq.(12) in the case of the quadratic min-max game of Eq.(13), using spectral analysis,

$$\boldsymbol{x}_{k+1} = \boldsymbol{x}_k + \beta\Delta\boldsymbol{x}_k - \eta h\boldsymbol{x}_k - \eta\mathbb{A}\boldsymbol{y}_k - \alpha\mathbb{A}\Delta\boldsymbol{y}_k$$
$$\boldsymbol{y}_{k+1} = \boldsymbol{y}_k + \beta\Delta\boldsymbol{y}_k - \eta h\boldsymbol{y}_k + \eta\mathbb{A}^T\boldsymbol{x}_k + \alpha\mathbb{A}^T\Delta\boldsymbol{x}_k, \tag{18}$$

where $\Delta\boldsymbol{x}_k = \boldsymbol{x}_k - \boldsymbol{x}_{k-1}$.

For brevity, consider the joint parameters $\boldsymbol{\omega}_t := (\boldsymbol{x}_t, \boldsymbol{y}_t)$. We start by studying the update operator of simultaneous gradient descent-ascent,

$$F_\eta(\boldsymbol{\omega}_t) = \boldsymbol{\omega}_t - \eta\boldsymbol{v}(\boldsymbol{\omega}_{t-1}).$$

where, the vector-field is given by Eq. 14. Thus, the fixed point $\boldsymbol{\omega}^*$ of $F_\eta(\boldsymbol{\omega}_t)$ satisfies $F_\eta(\boldsymbol{\omega}^*) = \boldsymbol{\omega}^*$.

Furthermore, at $\boldsymbol{\omega}^*$, we have,

$$\nabla F_\eta(\boldsymbol{\omega}^*) = \mathbb{I}_n - \eta \nabla \boldsymbol{v}(\boldsymbol{\omega}^*), \tag{19}$$

with $\mathbb{I}_n$ being the $n \times n$ identity matrix. Consequently the spectrum of $\nabla F_\eta(\boldsymbol{\omega}^*)$ in the quadratic game considered, is,

$$\mathrm{Sp}(\nabla F_\eta(\boldsymbol{\omega}^*)) = \{1 - \eta h - \eta \lambda \mid \lambda \in \mathrm{Sp}(\text{off-diag}[\nabla \boldsymbol{v}(\boldsymbol{\omega}^*)])\}, \tag{20}$$

The next proposition outlines the condition under which the fixed point operator is guaranteed to converge around the fixed point.

**Proposition 2 (Prop. 4.4.1 (Bertsekas, 1999)).** *For the spectral radius,*

$$\rho_{max} := \rho\{\nabla F_\eta(\boldsymbol{\omega}^*)\} < 1 \tag{21}$$

*and for some $\boldsymbol{\omega}_0$ in a neighborhood of $\boldsymbol{\omega}^*$, the update operator $F$, ensures linear convergence to $\boldsymbol{\omega}^*$ at a rate,*

$$\Delta_{t+1} \leq \mathcal{O}(\rho + \epsilon)\Delta_t \ \forall \ \epsilon > 0,$$

*where $\Delta_{t+1} := ||\boldsymbol{\omega}_{t+1} - \boldsymbol{\omega}^*||_2^2 + ||\boldsymbol{\omega}_t - \boldsymbol{\omega}^*||_2^2$.*

Next, we proceed to define the update operator of Eq.(12) as $F_{\mathrm{LEAD}}(\boldsymbol{\omega}_t, \boldsymbol{\omega}_{t-1}) = (\boldsymbol{\omega}_{t+1}, \boldsymbol{\omega}_t)$ . For the quadratic min-max game of Eq.(13), the Jacobian of $F_{\mathrm{LEAD}}$ takes the form,

$$\nabla F_{\mathrm{LEAD}} = \begin{bmatrix} \mathbb{I}_{2n} + \beta \mathbb{I}_{2n} - (\eta + \alpha)\nabla \boldsymbol{v} & -\beta \mathbb{I}_{2n} + \alpha \nabla \boldsymbol{v} \\ \mathbb{I}_{2n} & 0 \end{bmatrix}. \tag{22}$$

In the next Theorem 2, we find the set of eigenvalues corresponding to the update operator $\nabla F_{\mathrm{LEAD}}$ which are then used in Theorem 3, where we show for a selected values of $\eta$ and $\alpha$, LEAD attains a linear rate.

**Theorem 2.** *The eigenvalues of $\nabla F_{LEAD}(\boldsymbol{\omega}^*)$ are,*

$$\mu_\pm = \frac{1 - (\eta + \alpha)\lambda + \beta - \eta h \pm \sqrt{\Delta}}{2} \tag{23}$$

*where,*

$$\Delta = (1 - (\eta + \alpha)\lambda + \beta - \eta h)^2 - 4(\beta - \alpha\lambda)$$

*and $\lambda \in Sp(\text{off-diag}[\nabla \boldsymbol{v}(\boldsymbol{\omega}^*)])$.*

*Furthermore, for $h, \eta, |\alpha|, |\beta| << 1$, we have,*

$$\begin{aligned} \mu_+ \approx &1 - \eta h + \frac{(\eta + \alpha)^2 \lambda^2 + \eta^2 h^2 + \beta^2 - 2\eta h \beta}{4} \\ &+ \lambda\left(\frac{\eta + \alpha}{2}(\eta h - \beta) - \eta\right) \end{aligned} \tag{24}$$

*and*

$$\begin{aligned} \mu_- \approx &\beta - \frac{(\eta + \alpha)^2 \lambda^2 + \eta^2 h^2 + \beta^2 - 2\eta h \beta}{4} \\ &+ \lambda\left(\frac{\eta + \alpha}{2}(\beta - \eta h) - \alpha\right) \end{aligned} \tag{25}$$

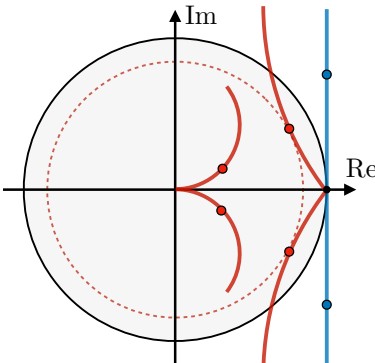

Figure 2: Diagram depicts positioning of the eigenvalues of GDA in blue (Eq. 19) and those of LEAD (eqs. (24) and (25) with $\beta = h = 0$) in red. Eigenvalues inside the black unit circle imply convergence such that the closer to the origin, the faster the convergence rate (Prop. 2). Every point on solid blue and red lines corresponds to a specific choice of learning rate. No choice of learning rate results in convergence for gradient ascent descent method as the blue line is tangent to the unit circle. At the same time, for a fixed value of $\alpha$, LEAD shifts the eigenvalues $(\mu_+)$ into the unit circle which leads to a convergence rate proportional to the radius of the red dashed circle. Note that LEAD also introduces an extra set of eigenvalues $(\mu_-)$ which are close to zero and do not affect convergence .

See proof in Appendix D.

Theorem 2 states that the LEAD operator has two eigenvalues $\mu_+$ and $\mu_-$ for each $\lambda \in \mathrm{Sp}\left(\text{off-diag}[\nabla \boldsymbol{v}(\boldsymbol{\omega}^*)]\right)$. Specifically, $\mu_+$ can be viewed as a shift of the eigenvalues of GDA in Eq.(20), while additionally being the leading eigenvalue for small values of $h, \eta, |\alpha|$ and $|\beta|$. (See Fig. 2 for a schematic description) Also, for small values of $\alpha$, $\mu_+$ is the limiting eigenvalue while $\mu_- \approx 0$.

In the following Proposition, we next show that locally, a choice of positive $\alpha$ decreases the spectral radius of $\nabla F_\eta (\boldsymbol{\omega}^*)$ defined as,

$$\rho := \max\{|\mu_+|^2, |\mu_-|^2\} \ \forall \ \lambda.$$

**Proposition 3.** *For any* $\lambda \in Sp(\text{off-diag}[\nabla \boldsymbol{v}(\omega^*)])$,

$$\nabla_\alpha \rho\left(\lambda\right)\big|_{\alpha=0} < 0 \Leftrightarrow \eta \in \left(0, \frac{2}{\mathrm{Im}(\lambda_{max})}\right), \tag{26}$$

*where* $\mathrm{Im}(\lambda_{max})$ *is the imaginary component of the largest eigenvalue* $\lambda_{max}$.

See proof in Appendix E.

Having established that a small positive value of $\alpha$ improves the rate of convergence, in the next theorem, we prove that for a specific choice of positive $\alpha$ and $\eta$ in the quadratic game Eq.(13), a linear rate of convergence to its Nash equilibrium is attained.

**Theorem 3.** *Setting* $\eta = \alpha = \frac{1}{2(\sigma_{max}(\mathbb{A})+h)}$*, then we have* $\forall \ \epsilon > 0$,

$$\Delta_{t+1} \in \mathcal{O}\left(\left(1 - \frac{\sigma_{min}^2}{4(\sigma_{max}+h)^2} - \frac{(1+\beta/2)h}{\sigma_{max}+h} + \frac{3h^2}{8(\sigma_{max}+h)^2}\right)^t \Delta_0\right) \tag{27}$$

*where* $\sigma_{max}(\sigma_{min})$ *is the largest (smallest) eigen value of* $\mathbb{A}$ *and*

$$\Delta_{t+1} := ||\boldsymbol{\omega}_{t+1} - \boldsymbol{\omega}^*||_2^2 + ||\boldsymbol{\omega}_t - \boldsymbol{\omega}^*||_2^2.$$

Theorem 3 ensures a linear convergence of LEAD in the quadratic min-max game. (Proof in Appendix F).

## 5 Comparison of Convergence Rate for Quadratic Min-Max Game

In this section, we perform a Big-O comparison of convergence rates of LEAD (Eq. 59), with several other existing methods. Below in Table 5 we summarize the convergence rates for the quadratic min-max game of Eq. 13. For each method that converges at the rate of $\mathcal{O}((1-r)^t)$, we report the quantity $r$ (larger $r$ corresponds to faster convergence). We observe that for the quadratic min-max game, given the analysis in Azizian et al. (2020a), for $h < \sigma_{\max}(\mathbb{A})$ and $\beta > \frac{3h^2}{8(\sigma_{max}+h)}$, $r_{\text{LEAD}} \gtrsim r_{\text{EG}}$ and $r_{\text{LEAD}} \gtrsim r_{\text{OG}}$. Furthermore, for the bilinear case, where $h = 0$, LEAD has a faster convergence rate than EG and OG.

| Method | r |
|---|---|
| Alternating-GDA | $h/2L$ |
| Extra-Gradient (EG) | $\frac{1}{4}(h/L + \sigma_{\min}^2(\mathbb{A})/16L^2)$ |
| Optimistic Gradient(OG) | $\frac{1}{4}(h/L + \sigma_{\min}^2(\mathbb{A})/32L^2)$ |
| Consensus Optimization (CO) | $h^2/2L_H^2 + \sigma_{\min}^2(\mathbb{A})/2L_H^2$ |
| LEAD (Th. 3) | $(1+\beta/2)h/(\sigma_{max}+h) + \sigma_{min}^2/4(\sigma_{max}+h)^2 - 3h^2/8(\sigma_{max}+h)^2$ |

Table 1: Big-O comparison of convergence rates of LEAD against EG (Korpelevich, 1976), OG (Mertikopoulos et al., 2018) and CO (Mescheder et al., 2017) for the **quadratic min-max game** of Eq. 13. We report the EG, OG and CO rates from the tight analysis in Azizian et al. (2020a) and Alt-GDA from Zhang et al. (2022). For each method that converges at the rate of $\mathcal{O}((1-r)^t)$, we report the quantity $r$ (larger $r$ corresponds to faster convergence). Note that $L := \sqrt{2}\max\{h, \sigma_{\max}(\mathbb{A})\}$, is the Lipschitz constant of the vector field and and $L_H^2$ is the Lipschitz-smoothness of $\frac{1}{2}\|v\|^2$.

## 6 Comparison of Computational Cost

In this section we first study several second-order algorithms and perform computational comparisons on an 8-Gaussians generation task. The Jacobian of the gradient vector field $\boldsymbol{v} = (\nabla_x f(\boldsymbol{x}, \boldsymbol{y}), -\nabla_y f(\boldsymbol{x}, \boldsymbol{y}))$ is given by,

$$\mathbb{J} = \begin{bmatrix} \nabla_x^2 f(\boldsymbol{x}, \boldsymbol{y}) & \nabla_{xy} f(\boldsymbol{x}, \boldsymbol{y}) \\ -\nabla_{yx} f(\boldsymbol{x}, \boldsymbol{y}) & -\nabla_y^2 f(\boldsymbol{x}, \boldsymbol{y}) \end{bmatrix}. \tag{28}$$

Considering player $\boldsymbol{x}$, a LEAD update requires the computation of the term $\nabla_{xy} f(\boldsymbol{x}_k, \boldsymbol{y}_k)(\boldsymbol{y}_k - \boldsymbol{y}_{k-1})$, thereby involving only one block of the full Jacobian $\mathbb{J}$. On the other hand Symplectic Gradient Adjustment (SGA) (Balduzzi et al., 2018a), requires the full computation of two Jacobian-vector products $\mathbb{J}\boldsymbol{v}, \mathbb{J}^\top \boldsymbol{v}$. Similarly, Competitive Gradient Descent (CGD) (Schäfer & Anandkumar, 2019) involves the computation of the following term,

$$\left(1 + \eta\nabla_{xy}^2 f(\boldsymbol{x}_k, \boldsymbol{y}_k)\nabla_{yx}^2 f(\boldsymbol{x}_k, \boldsymbol{y}_k)\right)^{-1}$$

along with the Jacobian-vector product,

$$\nabla_{xy}^2 f(\boldsymbol{x}_k, \boldsymbol{y}_k)\nabla_y f(\boldsymbol{x}_k, \boldsymbol{y}_k).$$

While the inverse term is approximated using conjugate gradient method, it still involves the computation of approximately ten Jacobian-vector products for each update. To explore these comparisons in greater detail and on models with many parameters, we experimentally compare the computational cost of our method with several other second as well as first-order methods on the 8-Gaussians problem in Figure 3 (architecture reported in Appendix J). We calculate the average wall-clock time (in milliseconds) per iteration. Results are reported on an average of 1000 iterations, computed on the same architecture and the same machine with forced synchronous execution. All the methods are implemented in PyTorch Paszke et al. (2017) and SGA is replicated based on the official implementation [2].

Furthermore, we observe that the computational cost per iteration of LEAD while being much lower than SGA and CGD, is similar to WGAN-GP and Extra-Gradient. The similarity to Extra-Gradient is due to

---

[2]SGA official DeepMind implementation (non-zero sum setting): https://github.com/deepmind/symplectic-gradient-adjustment/blob/master/Symplectic_Gradient_Adjustment.ipynb

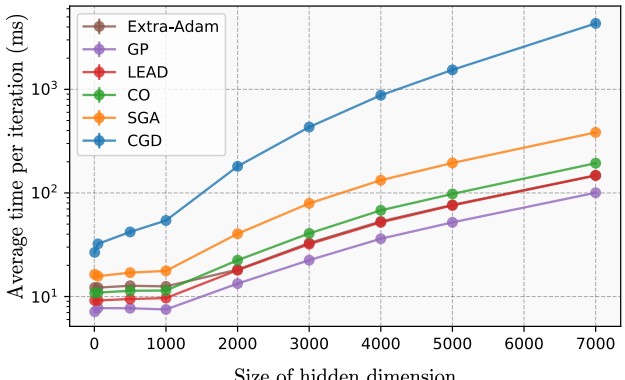

Figure 3: Average computational cost per iteration of several well-known methods for (non-saturating) GAN optimization. The numbers are reported on the 8-Gaussians generation task and averaged over 1000 iterations. Note that the y-axis is log-scale. We compare Competitive Gradient Descent (CGD) (53) (using official CGD optimizer code), Symplectic Gradient Adjustment (SGA) (6), Consensus Optimization (CO) (39), Extra-gradient with Adam (Extra-Adam) (17), WGAN with Gradient Penalty (WGAN GP) (21). We observe that per-iteration time complexity of our method is very similar to Extra-Adam and WGAN GP and is much cheaper than other second order methods such as CGD. Furthermore, by increasing the size of the hidden dimension of the generator and discriminator's networks we observe that the gap between different methods increases.

the fact that for each player, Extra-Gradient requires the computation of a half-step and a full-step, so in total each step requires the computation of two gradients. LEAD also requires the computation of a gradient $(\nabla f_x)$ which is then used to compute $(\nabla f_{xy})$ multiplied by $(\boldsymbol{y}_k - \boldsymbol{y}_{k-1})$. Using PyTorch, we do not require to compute $\nabla f_{xy}$ and then perform the multiplication. Given $\nabla f_x$ the whole term $\nabla f_{xy}(\boldsymbol{y}_k - \boldsymbol{y}_{k-1})$, is computed using PyTorch's Autograd Vector-Jacobian product, with the computational cost of a single gradient. Thus, LEAD also requires the computation of two gradients for each step.

## 7 Experiments

In this section, we first empirically validate the performance of LEAD on several toy as well as large-scale experiments. Furthermore, we extend LEAD based on the Adam algorithm to be used in large-scale experiments. See 2 for the detailed Algorithm.

### 7.1 Adversarial vs Cooperative Games

In Section 6 we showed that using Auto-grad software tools such as TensorFlow and PyTorch, LEAD can be computed very efficiently and as fast as extra-gradient. In this section we compare the perforamce of LEAD with several first order methods in a toy setup inspired by (Lorraine & Duvenaud, 2022). Consider the following game,

$$\min_{\boldsymbol{x}} \max_{\boldsymbol{y}} \boldsymbol{x}^T (\boldsymbol{\gamma A}) \boldsymbol{y} + \boldsymbol{x}^T ((\mathbf{I} - \boldsymbol{\gamma}) \boldsymbol{B}_1) \boldsymbol{x} - \boldsymbol{y}^T ((\mathbf{I} - \boldsymbol{\gamma}) \boldsymbol{B}_2) \boldsymbol{y}. \tag{29}$$

Such formulation in Eq. 29 enables us to compare the performance of different methods in cooperative games, adversarial games, and any interpolation between the two. Namely, varying $\gamma$ from $\mathbf{0}$ to $\mathbf{I}$ changes the dynamics of the game from purely cooperative to adversarial. Many real-world applications such as GANs exhibit an analogous range of adversarial to cooperative changes during training (Lorraine & Duvenaud, 2022).

In Figure 4, we compare LEAD against several methods including gradient descent-ascent (GDA), extragradient (EG) (Korpelevich, 1976), optimistic gradient (OG) (Mertikopoulos et al., 2018), complex momentum (CM) (Lorraine & Duvenaud, 2022), negative momentum (NM) (Gidel et al., 2019), and positive momentum (PM) (Polyak, 1964). Each method is tuned to optimality for each setup.

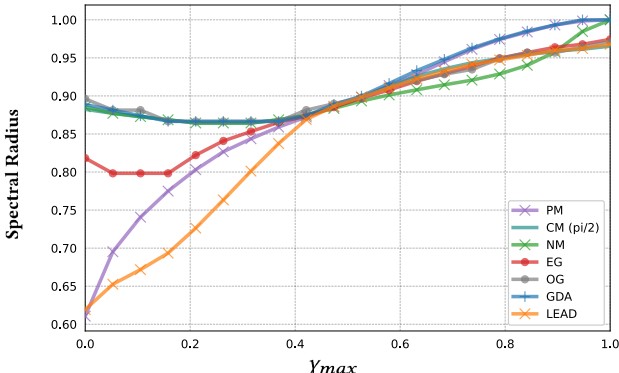

Figure 4: Comparison of several methods on the game in Eq. 29. The diagonal matrix $\gamma$ determines the degree of adversarialness along each direction. Elements on the diagonal are sampled from a uniform distribution, $\gamma_{ii} \sim \text{Unif}[0, \gamma_{max}]$. By varying $\gamma_{max}$ from 0 to 1, we move from a purely cooperative setup to a hybrid setup with a mixture of cooperative and adversarial games. The spectral radius (shown on the y-axis) determines the convergence rate in this game and is a function of $\gamma_{max}$. The smaller the spectral radius, the faster the convergence rate. A spectral radius of 1 corresponds to non-convergent dynamics. We compare several methods including gradient descent-ascent (GDA), extragradient (EG) (Korpelevich, 1976), optimistic gradient (OG) (Mertikopoulos et al., 2018), complex momentum (CM) (Lorraine & Duvenaud, 2022), negative momentum (NM) (Gidel et al., 2019), and positive momentum (PM) (Polyak, 1964). Each method has been tuned to optimality for each setup. For cooperative games (on the leftmost), LEAD and Positive Momentum achieve great performance. In more adversarial settings (on the rightmost), LEAD performs on par with other game-specific optimization methods (excluding Negative Momentum, GDA and Positive Momentum which diverge). This plot suggests that LEAD is a robust optimizer across different types of games. We conjecture that for the same reason, LEAD performs desirably in real-world setups such as GANs where the adversarialness changes dynamically throughout the training (Lorraine & Duvenaud, 2022).

## 7.2 Generative Adversarial Networks

We study the performance of LEAD in zero-sum as well as non-zero sum settings. See Appendix I for a comparison of LEAD-Adam against vanilla Adam on the generation task of a mixture of 8-Gaussians.

**CIFAR-10 DCGAN**: We evaluate LEAD-Adam on the task of CIFAR-10 (Krizhevsky & Hinton, 2009) image generation with a non-zero-sum formulation (non-saturating) on a DCGAN architecture similar to Gulrajani et al. (2017). As shown in Table 2, we compare with several first-order and second order methods and observe that LEAD-Adam outperforms the rest in terms of Fréchet Inception Distance (FID) (Heusel et al., 2017) and Inception score (IS) (Salimans et al., 2016) [3], reaching an FID score of 19.27±0.10 which outperforms OMD (Mertikopoulos et al., 2018) and CGD (Schäfer & Anandkumar, 2019). See Figure 5 that shows the improvement of FID using LEAD-Adam vs vanilla Adam and Figure 9 for a sample of the generated images.

**CIFAR-10 ResNet**: Furthermore, we evaluate LEAD-Adam on more complex and deep architectures. We adapt the ResNet architecture in SN-GAN Miyato et al. (2018). We compare with several existing results on the task of image generation on CIFAR-10 using ResNets. See Table 2 for a full comparison. Note that state of the art performance in recent work such as Style-GAN based models (Sauer et al., 2022; Kang et al., 2021; Lee et al., 2021) or BigGAN based models (Brock et al., 2018; Lorraine & Duvenaud, 2022) use architectures that are 30 times or more larger than the architecture that we have chosen to test our method on.

We report our results against a properly tuned version of SNGAN that achieves an FID of 12.36. Our method obtains a competitive FID of 10.49. We give a detailed description of these experiments and full detail on the architecture and hyper-parameters in Appendix J. See also Figure 10 for a sample of generated samples on a ResNet using LEAD-Adam.

---

[3]The FID and IS are metrics for evaluating the quality of generated samples of a generative model. Lower FID and higher inception score (IS) correspond to better sample quality.

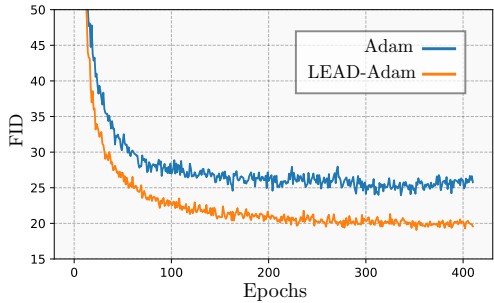

Figure 5: Plot showing the evolution of the FID over 400 epochs for our method (LEAD-Adam) vs vanilla Adam on a DCGAN architecture. It is important to note that compared to the Adam, LEAD-Adam is twice expensive computationally.

Table 2: Performance of several methods on CIFAR-10 image generation task. The FID and IS is reported over 50k samples unless mentioned otherwise.

| **DCGAN** | **FID** ($\downarrow$) | **IS** ($\uparrow$) |
|---|---|---|
| Adam (Radford et al., 2015) | $24.38 \pm 0.13$ | 6.58 |
| LEAD-Adam | $19.27 \pm 0.10$ | $7.58 \pm 0.11$ |
| CGD-WGAN (Schäfer & Anandkumar, 2019) | 21.3 | 7.2 |
| OMD (Daskalakis et al., 2018) | $29.6 \pm 0.19$ | $5.74 \pm 0.1$ |
| **ResNet** | | |
| SNGAN | $12.10 \pm 0.31$ | $8.58 \pm 0.03$ |
| LEAD-Adam (ours) | $\mathbf{10.49} \pm 0.11$ | $\mathbf{8.82} \pm 0.05$ |
| ExtraAdam (Gidel et al., 2018) | $16.78 \pm 0.21$ | $8.47 \pm 0.1$ |
| LA-GAN (Chavdarova et al., 2020) | $12.67 \pm 0.57$ | $8.55 \pm 0.04$ |
| ODE-GAN (Qin et al., 2020) | $11.85 \pm 0.21$ | $8.61 \pm 0.06$ |
| **Evaluated with 5k samples** | | |
| SN-GAN (DCGAN) (Miyato et al., 2018) | 29.3 | $7.42 \pm 0.08$ |
| SN-GAN (ResNet) (Miyato et al., 2018) | $21.7 \pm 0.21$ | $8.22 \pm 0.05$ |

# 8 Related Work

**Game Optimization**: With increasing interest in games, significant effort is being spent in understanding common issues affecting optimization in this domain. These issues range from convergence to non-Nash equilibrium points, to exhibiting rotational dynamics around the equilibrium which hampers convergence. Authors in Mescheder et al. (2017) discuss how the eigenvalues of the Jacobian govern the local convergence properties of GANs. They argue that the presence of eigenvalues with zero real-part and large imaginary part results in oscillatory behavior. To mitigate this issue, they propose Consensus Optimization (CO). Along similar lines, Balduzzi et al. (2018b); Gemp & Mahadevan (2018); Letcher et al. (2019); Loizou et al. (2020) use the *Hamiltonian* of the gradient vector-field, to improve the convergence in games through disentangling the convergent parts of the dynamics from the rotations. Another line of attack taken in Schäfer & Anandkumar (2019) is to use second-order information as a regularizer of the dynamics and motivate the use of Competitive Gradient Descent (CGD). In Wang et al. (2019), Follow the Ridge (FtR) is proposed. They motivate the use of a second order term for one of the players (follower) as to avoid the rotational dynamics in a sequential formulation of the zero-sum game. See appendix K for full discussion on the comparison of LEAD versus other second-order methods.

Another approach taken by Gidel et al. (2019), demonstrate how applying negative momentum over GDA can improve convergence in min-max games, while also proving a linear rate of convergence in the case of bilinear games. More recently, Zhang & Wang (2021) have shown the suboptimality of negative momentum in specific settings. Furthermore, in (Lorraine & Duvenaud, 2022) authors carry-out an extensive study on the effect of momentum in games and specifically show that complex momentum is optimal in many games ranging from adversarial to non-adversarial settings. Daskalakis et al. (2018) show that extrapolating the next value of the gradient using previous history, aids convergence. In the same spirit, Chavdarova et al. (2020), proposes LookAhead GAN (LA-GAN) and show that the LookAhead algorithm is a compelling candidate in improving

convergence in GANs. Gidel et al. (2018) also explores this line of thought by introducing averaging to develop a variant of the extra-gradient algorithm and proposes Extra-Adam-Averaging. Similar to Extra-Adam-Averaging is SN-EMA Yazıcı et al. (2019) which uses the SN-GAN and achieves great performance by applying an exponential moving average on the parameters. More recently, Fiez & Ratliff (2021) study using different time-scales for each player in zero-sum non-convex, non-concave games. Furthermore, in Rosca et al. (2021) authors study the dynamics of game optimization in both continuous and discrete time and examine the effects of discritization drift on the game performance. They suggest a modified continuous-time dynamical system that more closely matches the discrete time dynamics and introduce regularizers that mitigate the effect harmful drifts.

Lastly, in regard to convergence analysis in games, Zhang et al. (2022) study the convergence of alternating gradient descent-ascent for minmax games, Golowich et al. (2020) provide last iterate convergence rate for convex-concave saddle point problems. Nouiehed et al. (2019) propose a multi-step variant of gradient descent-ascent, to show it can find a game's $\epsilon$–first-order stationary point. Additionally, Azizian et al. (2020a) and Ibrahim et al. (2020) provide spectral lower bounds for the rate of convergence in the bilinear setting for an accelerated algorithm developed in Azizian et al. (2020b) for a specific families of bilinear games. Furthermore, Fiez & Ratliff (2020) use Lyapunov analysis to provide convergence guarantees for gradient descent ascent using timescale separation and in Hsieh et al. (2020), authors show that commonly used algorithms for min-max optimization converge to attractors that are not optimal.

**Single-objective Optimization and Dynamical Systems**: The authors of Su et al. (2014) started a new trend in single-objective optimization by studying the continuous-time dynamics of Nesterov's accelerated method (Nesterov, 2013). Their analysis allowed for a better understanding of the much-celebrated Nesterov's method. In a similar spirit, Wibisono et al. (2016); Wilson et al. (2016) study continuous-time accelerated methods within a Lagrangian framework, while analyzing their stability using Lyapunov analysis. These work show that a family of discrete-time methods can be derived from their corresponding continuous-time formalism using various discretization schemes. Additionally, several recent work (Muehlebach & Jordan, 2019; Bailey & Piliouras, 2019; Maddison et al., 2018; Ryu et al., 2019) cast game optimization algorithms as dynamical systems so to leverage its rich theory, to study the stability and convergence of various continuous-time methods. Nagarajan & Kolter (2017) also analyzes the local stability of GANs as an approximated continuous dynamical system.

## 9 Conclusion and Future Direction

In this paper, we leverage tools from physics to propose a novel second-order optimization scheme LEAD, to address the issue of rotational dynamics in min-max games. By casting min-max game optimization as a physical system, we use the principle of least action to discover an effective optimization algorithm for this setting. Subsequently, with the use of Lyapunov stability theory and spectral analysis, we prove LEAD to be convergent at a linear rate in bilinear min-max games. We supplement our theoretical analysis with experiments on GANs and toy setups, demonstrating improvements over baseline methods. Specifically for GAN training, we observe that our method outperforms other second-order methods, both in terms of sample quality and computational efficiency. Our analysis underlines the advantages of physical approaches in designing novel optimization algorithms for games as well as for traditional optimization tasks.

It is important to note in this regard that our crafted physical system is *a* way to model min-max optimization physically. Alternate schemes to perform such modeling can involve other choices of counter-rotational and dissipative forces which can be explored in future work. Other directions for future work can extend the Lyapunov analysis to the general convex-concave setting. Our existent analysis and the subsequent promising experimental results there upon, makes us believe that analysis can be extended to the more general convex-concave setting. Nevertheless, such an attempt is dependent on finding an appropriate Lyapunov function which may be challenging given the complex game dynamics.

As a future direction, one may pursue other types of games where LEAD may be useful. Any 2-parameter dynamical system can be interpreted as a 2-player game. Since LEAD models second-order interactions of the players, it may be a preferable optimization algorithm for games with higher-order structure. Furthermore,

studying the performance of LEAD on larger GAN architecture such as Style-GAN may be studied in future work.

## Broader Impact Statement

While our contribution is mostly theoretical, our research has the potential to improve the optimization of multi-agent machine learning models, such as generative adversarial networks (GANs). GANs have been very successful in generating realistic images, music, speech and text, and for improving performance on an array of different real-world tasks. On the other hand, GANs can be misused to generate fake news, fake images, and fake voices. Furthermore, a common problem encountered during GAN training is mode-collapse. This results in GANs being biased in generating certain types of data moreover others, thereby causing data misrepresentation. In this paper, we show that our proposed method can tackle the mode collapse problem by observing improvements over baseline methods. However, we would like to emphasize that practitioners should use our research with caution as the change of dataset and tasks might not prevent the mode collapse problem.

## Acknowledgments

The authors would like to thank Mohammad Pezeshki, Gauthier Gidel, Tatjana Chavdarova, Maxime Laborde, Nicolas Loizou, Hugo Berard, Giancarlo Kerg, Manuela Girotti, Adam Ibrahim, Damien Scieur and Michael Mulligan, for useful discussions and feedback.

This work is supported by NSERC Discovery Grants (RGPIN-2018-04821 and RGPIN-2019-06512), FRQNT Young Investigator Startup Grants (2019-NC-253251 and 2019-NC-257943), a startup grant by IVADO, the Canada CIFAR AI chairs program and a collaborative grant from Samsung Electronics Co., Ltd.

Reyhane Askari Hemmat also acknowledges support of Borealis AI Graduate Fellowship, Microsoft Diversity Award and the NSERC Postgraduate Scholarships. Amartya Mitra acknowledges support of the internship program at Mila and the UCR Graduate Fellowship.

This research was enabled in part by compute resources, software and technical help provided by Mila (mila.quebec) and Compute Canada.

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

## A  Appendix

## B  Proof of Proposition 1

*Proof.* The EOMs of the quadratic game in continuous-time (Eq.(10)), can be discretized in using a combination of implicit and explicit update steps as (Shi et al., 2019),

$$x_{k+1} - x_k = \delta v_{k+1}^x, \tag{30a}$$

$$y_{k+1} - y_k = \delta v_{k+1}^y, \tag{30b}$$

$$v_{k+1}^x - v_k^x = -q\delta \nabla_{xy} f(x_k, y_k) v_k^y - \mu \delta v_k^x - \delta \nabla_x f(x_k, y_k) \tag{30c}$$

$$v_{k+1}^y - v_k^y = q\delta \nabla_{xy} f(x_k, y_k) v_k^x - \mu \delta v_k^y + \delta \nabla_y f(x_k, y_k) \tag{30d}$$

where $\delta$ is the discretization step-size. Using Eqns.(30a) and (30b), we can further re-express Eqns. (30c), (30d) as,

$$\begin{aligned} x_{k+1} &= x_k + \beta \Delta x_k - \eta \nabla_x f(x_k, y_k) - \alpha \nabla_{x,y} f(x_k, y_k) \Delta y_k \\ y_{k+1} &= y_k + \beta \Delta y_k + \eta \nabla_y f(x_k, y_k) + \alpha \nabla_{x,y} f(x_k, y_k) \Delta x_k \end{aligned} \tag{31}$$

where $\Delta x_k = x_k - x_{k-1}$, and,

$$\beta = 1 - \mu\delta, \ \eta = \delta^2, \ \alpha = 2q\delta \tag{32}$$

$\square$

## C  Continuous-time Convergence Analysis: Quadratic Min-Max Game

*Proof.* For the class of quadratic min-max games,

$$f(\boldsymbol{X}, \boldsymbol{Y}) = \frac{h}{2}|\boldsymbol{X}|^2 - \frac{h}{2}|\boldsymbol{Y}|^2 + \boldsymbol{X}^T \mathbb{A} \boldsymbol{Y} \tag{33}$$

where $\boldsymbol{X} \equiv (X^1, \cdots, X^n), \boldsymbol{Y} \equiv (Y^1, \cdots, Y^n) \in \mathbb{R}^n$ and $\mathbb{A}_{n \times n}$ is a constant positive-definite matrix, the continuous-time EOMs of Eq.(10) become:

$$\begin{aligned} \ddot{\boldsymbol{X}} &= -\mu \dot{\boldsymbol{X}} - h\boldsymbol{X} - \mathbb{A}\boldsymbol{Y} - q\mathbb{A}\dot{\boldsymbol{Y}} \\ \ddot{\boldsymbol{Y}} &= -\mu \dot{\boldsymbol{Y}} - h\boldsymbol{Y} + \mathbb{A}^T \boldsymbol{X} + q\mathbb{A}^T \dot{\boldsymbol{X}} \end{aligned} \tag{34}$$

We next define our continuous-time Lyapunov function in this case to be,

$$\begin{aligned} \mathcal{E}_t &= \frac{1}{2} \left( \dot{\boldsymbol{X}} + \mu \boldsymbol{X} + \mu \mathbb{A} \boldsymbol{Y} \right)^T \left( \dot{\boldsymbol{X}} + \mu \boldsymbol{X} + \mu \mathbb{A} \boldsymbol{Y} \right) \\ &\quad + \frac{1}{2} \left( \dot{\boldsymbol{Y}} + \mu \boldsymbol{Y} - \mu \mathbb{A}^T \boldsymbol{X} \right)^T \left( \dot{\boldsymbol{X}} + \mu \boldsymbol{Y} - \mu \mathbb{A}^T \boldsymbol{X} \right) \\ &\quad + \frac{1}{2} \left( \dot{\boldsymbol{X}}^T \dot{\boldsymbol{X}} + \dot{\boldsymbol{Y}}^T \dot{\boldsymbol{Y}} \right) + \boldsymbol{X}^T (h + \mathbb{A}\mathbb{A}^T) \boldsymbol{X} + \boldsymbol{Y}^T (h + \mathbb{A}^T \mathbb{A}) \boldsymbol{Y} \\ &\geq 0 \ \forall \ t \end{aligned} \tag{35}$$

The time-derivative of $\mathcal{E}_t$ is then given by,

$$\begin{aligned} \dot{\mathcal{E}}_t &= \left( \dot{\boldsymbol{X}} + \mu \boldsymbol{X} + \mu \mathbb{A} \boldsymbol{Y} \right)^T \left( \ddot{\boldsymbol{X}} + \mu \dot{\boldsymbol{X}} + \mu \mathbb{A} \dot{\boldsymbol{Y}} \right) + \left( \dot{\boldsymbol{Y}} + \mu \boldsymbol{Y} - \mu \mathbb{A}^T \boldsymbol{X} \right)^T \left( \ddot{\boldsymbol{Y}} + \mu \dot{\boldsymbol{Y}} - \mu \mathbb{A}^T \dot{\boldsymbol{X}} \right) \\ &\quad + \left( \dot{\boldsymbol{X}}^T \ddot{\boldsymbol{X}} + \dot{\boldsymbol{Y}}^T \ddot{\boldsymbol{Y}} \right) + 2 \left( \boldsymbol{X}^T (h + \mathbb{A}\mathbb{A}^T) \dot{\boldsymbol{X}} + \boldsymbol{Y}^T (h + \mathbb{A}^T \mathbb{A}) \dot{\boldsymbol{Y}} \right) \\ &= \left( \dot{\boldsymbol{X}}^T + \mu \boldsymbol{X}^T + \mu \boldsymbol{Y}^T \mathbb{A}^T \right) \left( (-q + \mu) \mathbb{A} \dot{\boldsymbol{Y}} - \mathbb{A} \boldsymbol{Y} \right) + \dot{\boldsymbol{X}}^T \left( -q\mathbb{A}\dot{\boldsymbol{Y}} - \mu \dot{\boldsymbol{X}} - \mathbb{A}\boldsymbol{Y} \right) \\ &\quad + \left( \dot{\boldsymbol{Y}}^T + \mu \boldsymbol{Y}^T - \mu \boldsymbol{X}^T \mathbb{A} \right) \left( (q - \mu) \mathbb{A}^T \dot{\boldsymbol{X}} + \mathbb{A}^T \boldsymbol{X} \right) + \dot{\boldsymbol{Y}}^T \left( q\mathbb{A}^T \dot{\boldsymbol{X}} - \mu \dot{\boldsymbol{Y}} + \mathbb{A}^T \boldsymbol{X} \right) \\ &\quad + 2 \left( \boldsymbol{X}^T (h + \mathbb{A}\mathbb{A}^T) \dot{\boldsymbol{X}} + \boldsymbol{Y}^T (h + \mathbb{A}^T \mathbb{A}) \dot{\boldsymbol{Y}} \right) \\ &= (\mu(q - \mu) - 2) \left( \boldsymbol{Y}^T \mathbb{A}^T \dot{\boldsymbol{X}} - \boldsymbol{X}^T \mathbb{A} \dot{\boldsymbol{Y}} \right) - (\mu(q - \mu) - 2) \left( \boldsymbol{X}^T \mathbb{A}\mathbb{A}^T \dot{\boldsymbol{X}} + \boldsymbol{Y}^T \mathbb{A}^T \mathbb{A} \dot{\boldsymbol{Y}} \right) \\ &\quad - \mu \left( \boldsymbol{X}^T (h + \mathbb{A}\mathbb{A}^T) \boldsymbol{X} + \boldsymbol{Y}^T (h + \mathbb{A}^T \mathbb{A}) \boldsymbol{Y} \right) - \mu \left( \dot{\boldsymbol{X}}^T \dot{\boldsymbol{X}} + \dot{\boldsymbol{Y}}^T \dot{\boldsymbol{Y}} \right) \end{aligned} \tag{36}$$

where we have used the fact that $\boldsymbol{X}^T \mathbb{A} \boldsymbol{Y}$ being a scalar thus implying $\boldsymbol{X}^T \mathbb{A} \boldsymbol{Y} = \boldsymbol{Y}^T \mathbb{A}^T \boldsymbol{X}$. If we now set $q = (2/\mu) + \mu$ in the above, then that further leads to,

$$
\begin{aligned}
\dot{\mathcal{E}}_t &= -\mu \left( \boldsymbol{X}^T (h + \mathbb{A}\mathbb{A}^T) \boldsymbol{X} + \boldsymbol{Y}^T (h + \mathbb{A}^T \mathbb{A}) \boldsymbol{Y} \right) - \mu \left( \dot{\boldsymbol{X}}^T \dot{\boldsymbol{X}} + \dot{\boldsymbol{Y}}^T \dot{\boldsymbol{Y}} \right) \\
&= -\mu \left( h||\boldsymbol{X}||^2 + h||\boldsymbol{Y}||^2 + ||\mathbb{A}^T \boldsymbol{X}||^2 + ||\mathbb{A}\boldsymbol{Y}||^2 \right) - \mu \left( ||\dot{\boldsymbol{X}}||^2 + ||\dot{\boldsymbol{Y}}||^2 \right) \leq 0 \; \forall \; t
\end{aligned}
\tag{37}
$$

exhibiting that the Lyapunov function, Eq.(16) is *asymptotically stable* at all times $t$.

Next, consider the following expression,

$$
\begin{aligned}
&- \rho \mathcal{E}_t - \frac{\rho \mu}{2} ||\boldsymbol{X} - \dot{\boldsymbol{X}}||^2 - \frac{\rho \mu}{2} ||\boldsymbol{Y} - \dot{\boldsymbol{Y}}||^2 - \frac{\rho \mu}{2} ||\dot{\boldsymbol{X}} - \mathbb{A}\boldsymbol{Y}||^2 - \frac{\rho \mu}{2} ||\mathbb{A}^T \boldsymbol{X} + \dot{\boldsymbol{Y}}||^2 \\
&= -\rho \mathcal{E}_t - \frac{\rho \mu}{2} \left( ||\boldsymbol{X}||^2 + ||\boldsymbol{Y}||^2 \right) + \rho \mu \left( \boldsymbol{X}^T \dot{\boldsymbol{X}} + \boldsymbol{Y}^T \dot{\boldsymbol{Y}} \right) - \rho \mu \left( ||\dot{\boldsymbol{X}}||^2 + ||\boldsymbol{Y}||^2 \right) \\
&\quad - \rho \mu \left( \boldsymbol{X}^T \mathbb{A} \dot{\boldsymbol{Y}} - \dot{\boldsymbol{X}}^T \mathbb{A} \boldsymbol{Y} \right) - \frac{\rho \mu}{2} \left( ||\mathbb{A}^T \boldsymbol{X}||^2 + ||\mathbb{A}\boldsymbol{Y}||^2 \right) \\
&= -\rho (1 + \mu) \left( ||\dot{\boldsymbol{X}}||^2 + ||\dot{\boldsymbol{Y}}||^2 \right) - \frac{\rho}{2} \left( \mu^2 + \mu + 2h \right) \left( ||\boldsymbol{X}||^2 + ||\boldsymbol{Y}||^2 \right) \\
&\quad - \frac{\rho}{2} \left( \mu^2 + \mu + 2 \right) \left( ||\mathbb{A}^T \boldsymbol{X}||^2 + ||\mathbb{A}\boldsymbol{Y}||^2 \right) \\
&\leq -\rho \mathcal{E}_t
\end{aligned}
\tag{38}
$$

where $\rho$ is some positive definite constant. This implies that the above expression is negative semi-definite by construction given $\mu \geq 0$. Now, for a general square matrix $\mathbb{A}$, we can perform a singular value decomposition (SVD) as $\mathbb{A} = \mathbb{V}^T \mathbb{S} \mathbb{U}$. Here, $\mathbb{U}$ and $\mathbb{V}$ are the right and left unitaries of $\mathbb{A}$, while $\mathbb{S}$ is a diagonal matrix of singular values ($\sigma_i$) of $\mathbb{A}$. Using this decomposition in Eq.(38), then allows us to write,

$$
\begin{aligned}
&- \rho (1 + \mu) \left( ||\dot{\boldsymbol{X}}||^2 + ||\dot{\boldsymbol{Y}}||^2 \right) - \frac{\rho}{2} \left( \mu^2 + \mu + 2h \right) \left( ||\boldsymbol{X}||^2 + ||\boldsymbol{Y}||^2 \right) \\
&\quad - \frac{\rho}{2} \left( \mu^2 + \mu + 2 \right) \left( ||\mathbb{A}^T \boldsymbol{X}||^2 + ||\mathbb{A}\boldsymbol{y}||^2 \right) \\
&= -\rho (1 + \mu) \left( ||\mathbb{V}\dot{\boldsymbol{X}}||^2 + ||\mathbb{U}\dot{\boldsymbol{Y}}||^2 \right) - \frac{\rho}{2} \left( \mu^2 + \mu + 2h \right) \left( ||\mathbb{V}\boldsymbol{X}||^2 + ||\mathbb{U}\boldsymbol{Y}||^2 \right) \\
&\quad - \frac{\rho}{2} \left( \mu^2 + \mu + 2 \right) \left( ||\mathbb{S}\mathbb{V}\boldsymbol{X}||^2 + ||\mathbb{S}\mathbb{U}\boldsymbol{Y}||^2 \right) \\
&= -\rho (1 + \mu) \left( ||\dot{\boldsymbol{\mathcal{X}}}||^2 + ||\dot{\boldsymbol{\mathcal{Y}}}||^2 \right) - \frac{\rho}{2} \left( \mu^2 + \mu + 2h \right) \left( ||\boldsymbol{\mathcal{X}}||^2 + ||\boldsymbol{\mathcal{Y}}||^2 \right) \\
&\quad - \frac{\rho}{2} \left( \mu^2 + \mu + 2 \right) \left( ||\mathbb{S}\boldsymbol{\mathcal{X}}||^2 + ||\mathbb{S}\boldsymbol{\mathcal{Y}}||^2 \right) \\
&= -\sum_{j=1}^{n} \rho (1 + \mu) \left( ||\dot{\mathcal{X}}^j||^2 + ||\dot{\mathcal{Y}}^j||^2 \right) \\
&\quad - \sum_{j=1}^{n} \frac{\rho}{2} \left( (1 + \sigma_j^2 + 2h) (\mu^2 + \mu) + 2\sigma_j^2 \right) \left( ||\mathcal{X}^j||^2 + ||\mathcal{Y}^j||^2 \right)
\end{aligned}
\tag{39}
$$

where we have made use of the relations $\mathbb{U}^T \mathbb{U} = \mathbb{U}\mathbb{U}^T = \mathbb{I}_n = \mathbb{V}^T \mathbb{V} = \mathbb{V}\mathbb{V}^T$, and additionally performed a basis change, as $\boldsymbol{\mathcal{X}} = \mathbb{V}\boldsymbol{X}$ and $\boldsymbol{\mathcal{Y}} = \mathbb{U}\boldsymbol{Y}$. Now, we know from Eq.(37) that,

$$
\begin{aligned}
\dot{\mathcal{E}}_t &= -\mu \left( h||\boldsymbol{X}||^2 + h||\boldsymbol{Y}||^2 + ||\mathbb{A}^T \boldsymbol{X}||^2 + ||\mathbb{A}\boldsymbol{Y}||^2 \right) - \mu \left( ||\dot{\boldsymbol{X}}||^2 + ||\dot{\boldsymbol{Y}}||^2 \right) \\
&= -\mu \left( h||\boldsymbol{X}||^2 + h||\boldsymbol{Y}||^2 + ||\mathbb{U}^T \mathbb{S}\mathbb{V}\boldsymbol{X}||^2 + ||\mathbb{V}^T \mathbb{S}\mathbb{U}\boldsymbol{Y}||^2 \right) - \mu \left( ||\mathbb{V}\dot{\boldsymbol{X}}||^2 + ||\mathbb{U}\dot{\boldsymbol{Y}}||^2 \right) \\
&= -\mu \left( h||\boldsymbol{\mathcal{X}}||^2 + h||\boldsymbol{\mathcal{Y}}||^2 + ||\mathbb{S}\boldsymbol{\mathcal{X}}||^2 + ||\mathbb{S}\boldsymbol{\mathcal{Y}}||^2 \right) - \mu \left( ||\dot{\boldsymbol{\mathcal{X}}}||^2 + ||\dot{\boldsymbol{\mathcal{Y}}}||^2 \right) \\
&= -\sum_{j=1}^{n} \mu \left( \sigma_j^2 + h \right) \left( ||\mathcal{X}^j||^2 + ||\mathcal{Y}^j||^2 \right) - \sum_{j=1}^{n} \mu \left( ||\dot{\mathcal{X}}^j||^2 + ||\dot{\mathcal{Y}}^j||^2 \right)
\end{aligned}
\tag{40}
$$

Comparing the above expression with Eq.(39), we note that a choice of $\rho$ as,

$$\rho \le \min \left\{ \frac{\mu}{1+\mu}, \ \frac{2\mu(\sigma_{\min}^2 + h)}{(1 + \sigma_{\min}^2 + 2h)(\mu^2 + \mu) + 2\sigma_{\min}^2} \right\} \ \forall \ j \in [1, n] \tag{41}$$

implies,

$$\begin{aligned}
\dot{\mathcal{E}}_t &\le -\rho \mathcal{E} \\
\Rightarrow \mathcal{E}_t &\le \mathcal{E}_0 \exp(-\rho t) \\
\Rightarrow X^T \left(h + \mathbb{A}\mathbb{A}^T\right) X + Y^T \left(h + \mathbb{A}^T \mathbb{A}\right) Y &\le \mathcal{E}_0 \exp(-\rho t) \\
\Rightarrow \mathcal{X}^T (h + \mathbb{S}^2) \mathcal{X} + \mathcal{Y}^T (h + \mathbb{S}^2) \mathcal{Y} &\le \mathcal{E}_0 \exp(-\rho t) \\
\Rightarrow \sum_{j=1}^{n} (h + \sigma_j^2) \left(||\mathcal{X}^j||^2 + ||\mathcal{Y}^j||^2\right) &\le \mathcal{E}_0 \exp(-\rho t) \\
\Rightarrow \sum_{j=1}^{n} (h + \sigma_j^2) \left(||X^j||^2 + ||Y^j||^2\right) &\le \mathcal{E}_0 \exp(-\rho t) \\
\therefore \ ||\boldsymbol{X}||^2 + ||\boldsymbol{Y}||^2 &\le \frac{\mathcal{E}_0}{h + \sigma_{\min}^2} \exp(-\rho t) \ \forall \ j
\end{aligned} \tag{42}$$

$\square$

Figure 6: **Left:** Contours of the Lyapunov function $\mathcal{E}_k$, Eq. (35) (black), and convergence trajectory of LEAD (red) in the quadratic min-max game (Eq.(34)) to the Nash equilibrium $(0,0)$. **Right:** The evolution of the discrete-time Lyapunov function of Eq. (35) over iteration, confirming $\mathcal{E}_k - \mathcal{E}_{k-1} \le 0 \ \forall \ k \in \mathbf{N}$.

# D   Proof of Theorem 2

**Theorem.** *The eigenvalues of $\nabla F_{LEAD}(\boldsymbol{\omega}^*)$ about the Nash equilibrium $\boldsymbol{\omega}^* = (x^*, y^*)$ of the quadratic min-max game are,*

$$\mu_{\pm}(\alpha, \beta, \eta) = \frac{1 - (\eta + \alpha)\lambda + \beta - \eta h \pm \sqrt{\Delta}}{2} \tag{43}$$

where, $\Delta = (1 - (\eta + \alpha)\lambda + \beta - \eta h)^2 - 4(\beta - \alpha\lambda)$ and $\lambda \in Sp(\text{off-diag}[\nabla \boldsymbol{v}(\boldsymbol{\omega}^*)])$. Furthermore, for $h, \eta, |\alpha|, |\beta| << 1$, we have,

$$
\begin{aligned}
\mu_+^{(i)}(\alpha, \beta, \eta) &\approx 1 - \eta h + \frac{(\eta + \alpha)^2 \lambda_i^2 + \eta^2 h^2 + \beta^2 - 2\eta h \beta}{4} \\
&\quad + \lambda_i \left( \frac{\eta + \alpha}{2}(\eta h - \beta) - \eta \right) \\
\mu_-^{(i)}(\alpha, \beta, \eta) &\approx \beta - \frac{(\eta + \alpha)^2 \lambda_i^2 + \eta^2 h^2 + \beta^2 - 2\eta h \beta}{4} \\
&\quad + \lambda_i \left( \frac{\eta + \alpha}{2}(\beta - \eta h) - \alpha \right)
\end{aligned}
\tag{44}
$$

*Proof.* For the quadratic game 33, the Jacobian of the vector field $\boldsymbol{v}$ is given by,

$$
\nabla \boldsymbol{v} \equiv \nabla \begin{bmatrix} \nabla_x f(\boldsymbol{x}_t, \boldsymbol{y}_t) \\ -\nabla_y f(\boldsymbol{x}_t, \boldsymbol{y}_t) \end{bmatrix} = \begin{bmatrix} h\mathbb{I}_{2n} & \mathbb{A} \\ -\mathbb{A}^\top & h\mathbb{I}_{2n} \end{bmatrix} \in \mathbb{R}^{2n} \times \mathbb{R}^{2n}.
\tag{45}
$$

Let us next define a matrix $\mathbb{D}_q$ as,

$$
\mathbb{D}_q = \begin{bmatrix} \nabla_{xy}^2 f(\boldsymbol{x}, \boldsymbol{y}) & 0 \\ 0 & -\nabla_{xy}^2 f(\boldsymbol{x}, \boldsymbol{y}) \end{bmatrix} = \begin{bmatrix} \mathbb{A} & 0 \\ 0 & -\mathbb{A}^\top \end{bmatrix} \in \mathbb{R}^{2n} \times \mathbb{R}^{2n}
\tag{46}
$$

Consequently, the update rule for LEAD can be written as:

$$
\begin{aligned}
\begin{bmatrix} \boldsymbol{x}_{t+1} \\ \boldsymbol{y}_{t+1} \end{bmatrix} &= \begin{bmatrix} \boldsymbol{x}_t \\ \boldsymbol{y}_t \end{bmatrix} + \beta \begin{bmatrix} \boldsymbol{x}_t - \boldsymbol{x}_{t-1} \\ \boldsymbol{y}_t - \boldsymbol{y}_{t-1} \end{bmatrix} - \eta \begin{bmatrix} \nabla_x f(\boldsymbol{x}_t, \boldsymbol{y}_t) \\ -\nabla_y f(\boldsymbol{x}_t, \boldsymbol{y}_t) \end{bmatrix} - \alpha \begin{bmatrix} \nabla_{xy}^2 f(\boldsymbol{x}_t, \boldsymbol{y}_t) \Delta \boldsymbol{y}_t \\ -\nabla_{xy}^2 f(\boldsymbol{x}_t, \boldsymbol{y}_t) \Delta \boldsymbol{x}_t \end{bmatrix} \\
&= \begin{bmatrix} \boldsymbol{x}_t \\ \boldsymbol{y}_t \end{bmatrix} + \beta \begin{bmatrix} \boldsymbol{x}_t - \boldsymbol{x}_{t-1} \\ \boldsymbol{y}_t - \boldsymbol{y}_{t-1} \end{bmatrix} - \eta \boldsymbol{v} - \alpha \mathbb{D}_q \begin{bmatrix} \Delta \boldsymbol{y}_t \\ \Delta \boldsymbol{x}_t \end{bmatrix}
\end{aligned}
\tag{47}
$$

where $\Delta \boldsymbol{y}_t = \boldsymbol{y}_t - \boldsymbol{y}_{t-1}$ and $\Delta \boldsymbol{x}_t = \boldsymbol{x}_t - \boldsymbol{x}_{t-1}$.

Next, by making use of the permutation matrix $\mathbb{P}$,

$$
\mathbb{P} := \begin{bmatrix} 0 & \mathbb{I}_n \\ \mathbb{I}_n & 0 \end{bmatrix} \in \mathbb{R}^{2n} \times \mathbb{R}^{2n}
$$

we can re-express Eq. 47 as,

$$
\begin{aligned}
\begin{bmatrix} \boldsymbol{\omega}_{t+1} \\ \boldsymbol{\omega}_t \end{bmatrix} &= \begin{bmatrix} \mathbb{I}_{2n} & 0 \\ \mathbb{I}_{2n} & 0 \end{bmatrix} \begin{bmatrix} \boldsymbol{\omega}_t \\ \boldsymbol{\omega}_{t-1} \end{bmatrix} + \beta \begin{bmatrix} \mathbb{I}_{2n} & -\mathbb{I}_{2n} \\ 0 & 0 \end{bmatrix} \begin{bmatrix} \boldsymbol{\omega}_t \\ \boldsymbol{\omega}_{t-1} \end{bmatrix} - \eta \begin{bmatrix} \boldsymbol{v} \\ 0 \end{bmatrix} - \alpha \begin{bmatrix} \mathbb{D}_q & 0 \\ 0 & 0 \end{bmatrix} \begin{bmatrix} \mathbb{P} & -\mathbb{P} \\ 0 & 0 \end{bmatrix} \begin{bmatrix} \boldsymbol{\omega}_t \\ \boldsymbol{\omega}_{t-1} \end{bmatrix} \\
&= \begin{bmatrix} \mathbb{I}_{2n} & 0 \\ \mathbb{I}_{2n} & 0 \end{bmatrix} \begin{bmatrix} \boldsymbol{\omega}_t \\ \boldsymbol{\omega}_{t-1} \end{bmatrix} + \beta \begin{bmatrix} \mathbb{I}_{2n} & -\mathbb{I}_{2n} \\ 0 & 0 \end{bmatrix} \begin{bmatrix} \boldsymbol{\omega}_t \\ \boldsymbol{\omega}_{t-1} \end{bmatrix} - \eta \begin{bmatrix} \boldsymbol{v} \\ 0 \end{bmatrix} - \alpha \begin{bmatrix} \mathbb{D}_q \mathbb{P} & -\mathbb{D}_q \mathbb{P} \\ 0 & 0 \end{bmatrix} \begin{bmatrix} \boldsymbol{\omega}_t \\ \boldsymbol{\omega}_{t-1} \end{bmatrix}
\end{aligned}
\tag{48}
$$

where $\boldsymbol{\omega}_t \equiv (\boldsymbol{x}_t, \boldsymbol{y}_t)$. Hence, the Jacobian of $F_{\text{LEAD}}$ is then given by,

$$
\begin{aligned}
\nabla F_{\text{LEAD}} &= \begin{bmatrix} \mathbb{I}_{2n} & 0 \\ \mathbb{I}_{2n} & 0 \end{bmatrix} + \beta \begin{bmatrix} \mathbb{I}_{2n} & -\mathbb{I}_{2n} \\ 0 & 0 \end{bmatrix} - \eta \begin{bmatrix} \nabla \boldsymbol{v} & 0 \\ 0 & 0 \end{bmatrix} - \alpha \begin{bmatrix} \mathbb{D}_q \mathbb{P} & -\mathbb{D}_q \mathbb{P} \\ 0 & 0 \end{bmatrix} \\
&= \begin{bmatrix} (1 + \beta)\mathbb{I}_{2n} - \eta \nabla \boldsymbol{v} - \alpha \mathbb{D}_q \mathbb{P} & -\beta \mathbb{I}_{2n} + \alpha \mathbb{D}_q \mathbb{P} \\ \mathbb{I}_{2n} & 0 \end{bmatrix}
\end{aligned}
\tag{49}
$$

It is to be noted that, for games of the form of Eq. 33, we specifically have,

$$
\nabla \boldsymbol{v} = \mathbb{D}_q \mathbb{P} + h\mathbb{I}_{2n}
$$

and,

$$
\text{off-diag}[\nabla \boldsymbol{v}] = \mathbb{D}_q \mathbb{P}
$$

Therefore, Eq. 49 becomes,

$$\nabla F_{\text{LEAD}} = \begin{bmatrix} (1 + \beta - \eta h)\,\mathbb{I}_{2n} - (\eta + \alpha)\,\mathbb{D}_{\text{q}}\mathbb{P} & -\beta\mathbb{I}_{2n} + \alpha\mathbb{D}_{\text{q}}\mathbb{P} \\ \mathbb{I}_{2n} & 0 \end{bmatrix} \tag{50}$$

We next proceed to study the eigenvalues of this matrix which will determine the convergence properties of LEAD around the Nash equilibrium. Using Lemma 1 of (Gidel et al., 2019), we can then write the characteristic polynomial of $\nabla F_{\text{LEAD}}$ as,

$$
\begin{aligned}
&\det\left(X\mathbb{I}_{4n} - \nabla F_{\text{LEAD}}\right) = 0 \\
\Rightarrow &\det\left(\begin{bmatrix} (X-1)\,\mathbb{I}_{2n} - (\beta - \eta h)\,\mathbb{I}_{2n} + (\eta + \alpha)\,\mathbb{D}_{\text{q}}\mathbb{P} & \beta\mathbb{I}_{2n} - \alpha\mathbb{D}_{\text{q}}\mathbb{P} \\ -\mathbb{I}_{2n} & X\mathbb{I}_{2n} \end{bmatrix}\right) = 0 \\
\Rightarrow &\det\left(\left[(X-1)\,(X-\beta)\,\mathbb{I}_{2n} + X\eta h\mathbb{I}_{2n} + (X\eta + X\alpha - \alpha)\,\mathbb{D}_{\text{q}}\mathbb{P}\right]\right) = 0 \\
\Rightarrow &\det\left(\left[((X-1)\,(X-\beta) + X\eta h)\,\mathbf{U}\mathbf{U}^{-1} + (X\eta + X\alpha - \alpha)\,\mathbf{U}\mathbf{\Lambda}\mathbf{U}^{-1}\right]\right) = 0 \\
\Rightarrow &\det\left(\left[((X-1)\,(X-\beta) + X\eta h)\,\mathbb{I}_{2n} + (X\eta + X\alpha - \alpha)\,\mathbf{\Lambda}\right]\right) = 0 \\
\Rightarrow &\prod_{i=1}^{2n}\left[(X-1)\,(X-\beta) + X\eta h + (X\eta + \alpha\,(X-1))\,\lambda_i\right] = 0
\end{aligned}
\tag{51}
$$

Where, in the above, we have performed an eigenvalue decomposition of $\mathbb{D}_{\text{q}}\mathbb{P} = \mathbf{U}\mathbf{\Lambda}\mathbf{U}^{-1}$. Therefore,

$$
\begin{aligned}
&X^2 - X\left(1 - (\eta + \alpha)\lambda_i + \beta - \eta h\right) + \beta - \alpha\lambda = 0, \ \ \lambda_i \in \text{Sp}(\mathbb{D}_{\text{q}}\mathbb{P}) \\
\Rightarrow &X^{(i)} \equiv \mu_{\pm}^{(i)} = \frac{1 - (\eta + \alpha)\lambda_i + \beta - \eta h \pm \sqrt{\Delta}}{2}
\end{aligned}
\tag{52}
$$

with,

$$\Delta = \left(1 - (\eta + \alpha)\,\lambda_i + \beta - \eta h\right)^2 - 4\left(\beta - \alpha\lambda_i\right) \tag{53}$$

Furthermore for $h, \eta, |\beta|, |\alpha| << 1$, we can approximate the above roots to be,

$$
\begin{aligned}
\mu_{+}^{(i)}(\alpha, \beta, \eta) &\approx 1 - \eta h + \frac{(\eta + \alpha)^2\,\lambda_i^2 + \eta^2 h^2 + \beta^2 - 2\eta h\beta}{4} + \lambda_i\left(\frac{\eta + \alpha}{2}\,(\eta h - \beta) - \eta\right) \\
\mu_{-}^{(i)}(\alpha, \beta, \eta) &\approx \beta - \frac{(\eta + \alpha)^2\,\lambda_i^2 + \eta^2 h^2 + \beta^2 - 2\eta h\beta}{4} + \lambda_i\left(\frac{\eta + \alpha}{2}\,(\beta - \eta h) - \alpha\right)
\end{aligned}
\tag{54}
$$

$\square$

# E   Proof of Proposition 3

**Proposition.** *For any $\lambda \in Sp(\text{off-diag}[\nabla \boldsymbol{v}(\omega^*)])$,*

$$\nabla_{\alpha}\rho\left(\lambda\right)\big|_{\alpha=0} < 0 \Leftrightarrow \eta \in \left(0, \frac{2}{\text{Im}(\lambda_{max})}\right), \tag{55}$$

*where $\text{Im}(\lambda_{max})$ is the imaginary component of the largest eigenvalue $\lambda_{max}$.*

We observe from Proposition 3 above that for $h, \eta, |\alpha|, |\beta| << 1$,

$$
\begin{aligned}
\rho(\alpha, \eta, \beta) &:= \max\{|\mu_{+}^{(i)}|^2, |\mu_{-}^{(i)}|^2\} \ \forall \ i \\
&= \max\{\left|\mu_{+}^{(i)}\right|^2\} \ \forall \ i
\end{aligned}
\tag{56}
$$

$$\therefore \nabla_\alpha \rho\big|_{\alpha=0} \approx \max \left\{ \frac{\eta^2 |\lambda_i|^2 - \eta^2 h^2 - \beta^2}{4} \eta |\lambda_i|^2 + \frac{\eta h\beta - (\eta h - \beta)^2}{2} \eta |\lambda_i|^2 \right.$$
$$\left. - (1+\beta)\eta |\lambda_i|^2 \right\} \ \forall \ i$$
$$\approx \max \left\{ \frac{\eta^3}{4} |\lambda_i|^4 - \left(1 + \beta + \frac{3\beta^2}{4}\right) \eta |\lambda_i|^2 \right\} \ \forall \ i \tag{57}$$
$$< \max \left\{ \left(\frac{\eta^2}{4} |\lambda_i|^2 - 1\right) \eta |\lambda_i|^2 \right\} \ \forall \ i$$

where we have retained only terms up to cubic-order in $\eta, |\beta|$ and $h$. Hence, choosing $\eta \in \left(0, \frac{2}{\mathrm{Im}(\lambda_{\max})}\right)$, ensures:

$$\nabla_\alpha \rho\big|_{\alpha=0} < 0 \ \forall \ i, \tag{58}$$

We thus posit, that a choice of a positive $\alpha$ causes the norm of the limiting eigenvalue $\mu_+$ of $F_{\mathrm{LEAD}}$ to decrease.

## F  Proof of Theorem 3

**Theorem.** *Setting $\eta = \alpha = \frac{1}{2(\sigma_{max}(\mathbb{A})+h)}$, then we have $\forall \ \epsilon > 0$,*

$$\Delta_{t+1} \in \mathcal{O}\left(\left(1 - \frac{\sigma_{min}^2}{4(\sigma_{max}+h)^2} - \frac{(1+\beta/2)h}{\sigma_{max}+h} + \frac{3h^2}{8(\sigma_{max}+h)^2}\right)^t \Delta_0\right) \tag{59}$$

*where $\sigma_{max}(\sigma_{min})$ is the largest (smallest) eigenvalue of $\mathbb{A}$, $\Delta_{t+1} := ||\boldsymbol{\omega}_{t+1} - \boldsymbol{\omega}^*||_2^2 + ||\boldsymbol{\omega}_t - \boldsymbol{\omega}^*||_2^2$.*

*Proof*: From Eq. (52), we recall that the eigenvalues of $\nabla F_{\mathrm{LEAD}}(\boldsymbol{\omega}^*)$ for the quadratic game are,

$$\mu_\pm^{(i)}(\alpha, \beta, \eta) = \frac{(1 - (\alpha+\eta)\lambda_i + \beta - \eta h)}{2} \left(1 \pm \sqrt{1 - \frac{4(\beta - \eta\lambda_i)}{(1 - (\alpha+\eta)\lambda_i + \beta - \eta h)^2}}\right) \tag{60}$$

with $\lambda_i \in \mathrm{Sp}(\text{off-diag}[\nabla \boldsymbol{v}(\omega^*)])$. Now, since in the quadratic-game setting considered, we have,

$$\text{off-diag}[\nabla \boldsymbol{v}(\omega^*)] = \mathbb{D}_q \mathbb{P} = \begin{bmatrix} 0 & \mathbb{A} \\ -\mathbb{A}^T & 0 \end{bmatrix} \tag{61}$$

hence, $\lambda_i = \pm i\sigma_i$ with $\sigma_i$ being the singular values of $\mathbb{A}$. This, then allows us to write,

$$\mu_\pm^{(i)}(\alpha, \beta, \eta) = \frac{(1 - (\alpha+\eta)(\pm i\sigma_i) + \beta - \eta h)}{2} \left(1 \pm \sqrt{1 - \frac{4(\beta - \alpha(\pm i\sigma_i))}{(1 - (\alpha+\eta)(\pm i\sigma_i) + \beta - \eta h)^2}}\right) \tag{62}$$

According to Proposition 2, the convergence behavior of LEAD is determined as, $\Delta_{t+1} \le \mathcal{O}(\rho + \epsilon)\Delta_t \ \forall \ \epsilon > 0$, where (setting $\eta = \alpha$),

$$\rho := \max\{|\mu_+^{(i)}|^2, |\mu_-^{(i)}|^2\} \ \forall \ i$$
$$= |\mu_+^{(i)}|^2 \ \forall \ i \tag{63}$$

Now assuming that $\eta$ is small enough, such that, $\eta^3 \approx 0$ and $\beta^2 \approx 0$, we have,

$$\rho \approx 1 - \eta^2 \sigma_i^2 + \frac{3}{2}\eta^2 h^2 - (2+\beta)\eta h \tag{64}$$

Furthermore, using a learning rate $\eta$ as prescribed by Proposition 3, such as $\eta = \alpha = \frac{1}{2(\sigma_{\max}(\mathbb{A})+h)}$ we find,

$$r_{\mathrm{LEAD}} = \frac{\sigma_{min}^2}{4(\sigma_{max}+h)^2} + \frac{(1+\beta/2)h}{\sigma_{max}+h} - \frac{3h^2}{8(\sigma_{max}+h)^2} \tag{65}$$

Therefore,

$$
\begin{aligned}
\Delta_{t+1} &\leq \mathcal{O}\left((1 - r_{\text{LEAD}})^t \Delta_0\right) \\
&= \mathcal{O}\left(\left(1 - \frac{\sigma_{min}^2}{4(\sigma_{max} + h)^2} - \frac{(1 + \beta/2)h}{\sigma_{max} + h} + \frac{3h^2}{8(\sigma_{max} + h)^2}\right)^t \Delta_0\right)
\end{aligned}
\tag{66}
$$

where $\Delta_{t+1} := ||\boldsymbol{\omega}_{t+1} - \boldsymbol{\omega}^*||_2^2 + ||\boldsymbol{\omega}_t - \boldsymbol{\omega}^*||_2^2$.

## G    Robustness to the Discretization Parameter

In this section we study the effect of the discretization parameter $\delta$, defined in equation 32 and Proposition 1 on the quadratic game defined in 29,

$$
\min_{\boldsymbol{x}} \max_{\boldsymbol{y}} \boldsymbol{x}^T(\boldsymbol{\gamma A})\boldsymbol{y} + \boldsymbol{x}^T((\mathbf{I} - \boldsymbol{\gamma})\boldsymbol{B}_1)\boldsymbol{x} - \boldsymbol{y}^T((\mathbf{I} - \boldsymbol{\gamma})\boldsymbol{B}_2)\boldsymbol{y}.
$$

In Figure 7, we provide a grid of experiments to study the effect of discretization on the convergence of the quadratic game explored in section 7.1 (Adversarial vs Cooperative games). We observe that for every level of $\gamma_{max}$ that changes the dynamics of the game from adversarial to cooperative game, LEAD allows for a wide range of discretization steps that lead to convergence. Note that the discretization step and consequently the learning rate can be viewed as a hyper-parameter and consequently require tuning.

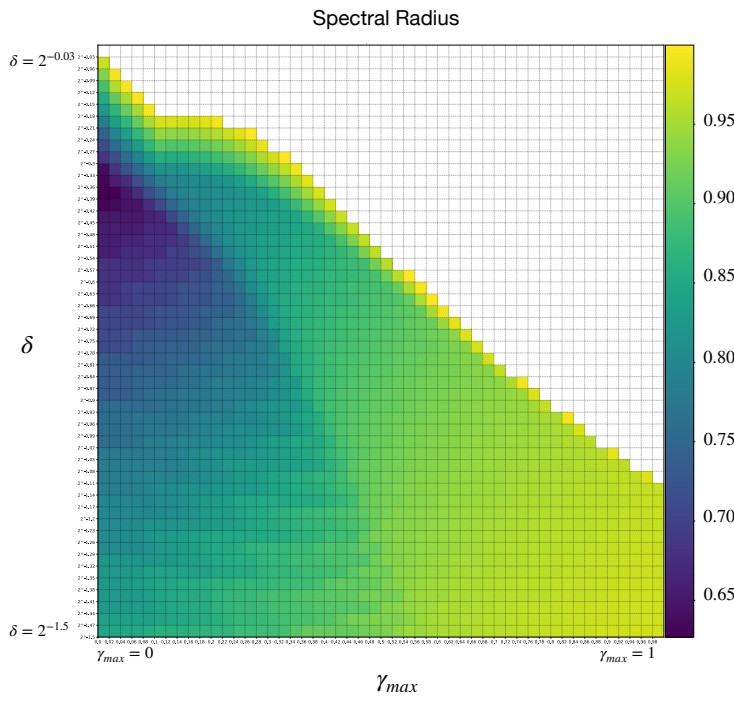

Figure 7: The effect of discretization parameter ($\delta$) on the convergence of LEAD on the quadratic min-max game defined in 29. The x-axis changes the game dynamics from cooperative to adversarial. The y-axes shows the range of $\delta$. The color shows the spectral radius which corresponds to the convergence rate (smaller spectral radius leads to faster convergence). Convergent dynamics, require a spectral radius that is smaller than one. We plot experiments with non-convergent dynamics (spectral radius $> 1$) in white. Darker colors correspond to faster convergence. We observe that LEAD allows for a wide range of $\delta$ that result in convergent dynamics. Thus, LEAD is robust against the discretization parameter.

## H    LEAD-Adam

Since Adam algorithm is commonly used in large-scale experiments, we extend LEAD to be used with the Adam algorithm.

---

**Algorithm 2** Least Action Dynamics Adam (LEAD-Adam)

---

1: **Input:** learning rate $\eta$, momentum $\beta$, coupling coefficient $\alpha$.
2: **Initialize:** $x_0 \leftarrow x_{init}$, $y_0 \leftarrow y_{init}$, $t \leftarrow 0$, $m_0^x \leftarrow 0$, $v_0^x \leftarrow 0$ $m_0^y \leftarrow 0$, $v_0^y \leftarrow 0$
3: **while** not converged **do**
4:     $t \leftarrow t + 1$
5:     $g_x \leftarrow \nabla_x f(x_t, y_t)$
6:     $g_{xy}\Delta y \leftarrow \nabla_y(g_x)(y_t - y_{t-1})$
7:     $g_t^x \leftarrow g_{xy}\Delta y + g_x$
8:     $m_t^x \leftarrow \beta_1 . m_{t-1}^x + (1 - \beta_1) . g_t^x$
9:     $v_t^x \leftarrow \beta_2 . v_{t-1}^x + (1 - \beta_2) . (g_t^x)^2$
10:     $\hat{m}_t \leftarrow m_t / (1 - \beta_1^t)$
11:     $\hat{v}_t \leftarrow v_t / (1 - \beta_2^t)$
12:     $x_{t+1} \leftarrow x_t - \eta \ \hat{m}_t / (\sqrt{\hat{v}_t} + \epsilon)$
13:     $g_y \leftarrow \nabla_y f(x_{t+1}, y_t)$
14:     $g_{xy}\Delta x \leftarrow \nabla_x(g_y)(x_{t+1} - x_t)$
15:     $g_t^y \leftarrow g_{xy}\Delta x + g_y$
16:     $m_t^y \leftarrow \beta_1 . m_{t-1}^y + (1 - \beta_1) . g_t^y$
17:     $v_t^y \leftarrow \beta_2 . v_{t-1}^y + (1 - \beta_2) . (g_t^y)^2$
18:     $\hat{m}_t^y \leftarrow m_t^y / (1 - \beta_1^t)$
19:     $\hat{v}_t^y \leftarrow v_t^y / (1 - \beta_2^t)$
20:     $y_{t+1} \leftarrow y_t + \eta \ \hat{m}_t^y / (\sqrt{\hat{v}_t^y} + \epsilon)$
21: **end while**
22: **return** $(x, y)$

---

# I   8-Gaussians Generation

We compare our method LEAD-Adam with vanilla-Adam (Kingma & Ba, 2014) on the generation task of a mixture of 8-Gaussians. Standard optimization algorithms such as vanilla-Adam suffer from mode collapse in this simple task, implying the generator cannot produce samples from one or several of the distributions present in the real data. Through Figure 8, we demonstrate that LEAD-Adam fully captures all the modes in the real data in both saturating and non-saturating losses.

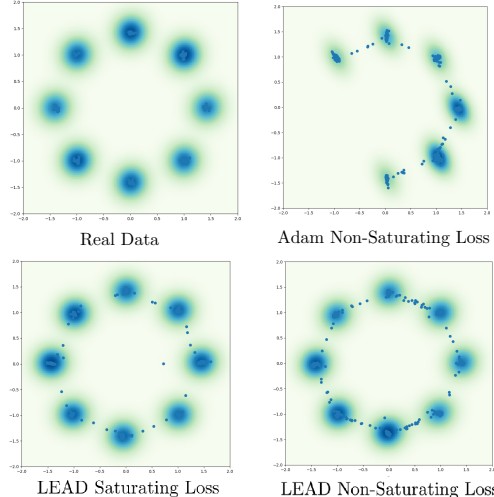

Figure 8: Performance of LEAD-Adam on the generation task of 8-Gaussians. All samples are shown after 10k iterations. Samples generated using Adam exhibit mode collapse, while LEAD-Adam does not suffer from this issue.

## J  Experiments and Implementation Details

### J.1  Mixture of Eight Gaussians

**Dataset** The real data is generated by 8-Gaussian distributions their mean are uniformly distributed around the unit circle and their variance is 0.05. The code to generate the data is included in the source code.

**Architecture** The architecture for Generator and Discriminator, each consists of four layers of affine transformation, followed by ReLU non-linearity. The weight initialization is default PyTorch's initialization scheme. See a schematic of the architecture in Table 3.

Table 3: Architecture used for the Mixture of Eight Gaussians.

| Generator | Discriminator |
|---|---|
| *Input: $z \in \mathbb{R}^{64} \sim \mathcal{N}(0, I)$* | *Input: $x \in \mathbb{R}^2$* |
| Linear $(64 \to 2000)$ | Linear $(2 \to 2000)$ |
| ReLU | ReLU |
| Linear $(2000 \to 2000)$ | Linear $(2000 \to 2000)$ |
| ReLU | ReLU |
| Linear $(2000 \to 2000)$ | Linear $(2000 \to 2000)$ |
| ReLU | ReLU |
| Linear $(2000 \to 2)$ | Linear $(2000 \to 1)$ |

**Other Details**  We use the Adam (Kingma & Ba, 2014) optimizer on top of our algorithm in the reported results. Furthermore, we use batchsize of 128.

### J.2  CIFAR 10 DCGAN

**Dataset** The CIFAR10 dataset is available for download at the following link; `https://www.cs.toronto.edu/~kriz/cifar.html`

**Architecture** The discriminator has four layers of convolution with LeakyReLU and batch normalization. Also, the generator has four layers of deconvolution with ReLU and batch normalization. See a schematic of the architecture in Table 4.

Table 4: Architecture used for CIFAR-10 DCGAN.

| Generator | Discriminator |
|---|---|
| *Input: $z \in \mathbb{R}^{100} \sim \mathcal{N}(0, I)$* | *Input: $x \in \mathbb{R}^{3 \times 32 \times 32}$* |
| conv. (ker: 4×4, $100 \to 1024$; stride: 1; pad: 0) | conv. (ker: 4×4, $3 \to 256$; stride: 2; pad: 1) |
| Batch Normalization | LeakyReLU |
| ReLU | conv. (ker: 4×4, $256 \to 512$; stride: 2; pad: 1) |
| conv. (ker: 4×4, $1024 \to 512$; stride: 2; pad: 1) | Batch Normalization |
| Batch Normalization | LeakyReLU |
| ReLU | conv. (ker: 4×4, $512 \to 1024$; stride: 2; pad: 1) |
| conv. (ker: 4×4, $512 \to 256$; stride: 2; pad: 1) | Batch Normalization |
| Batch Normalization | LeakyReLU |
| ReLU | conv. (ker: 4×4, $1024 \to 1$; stride: 1; pad: 0) |
| conv. (ker: 4×4, $256 \to 3$; stride: 2; pad: 1) | |
| Tanh | Sigmoid |

**Other Details** For the baseline we use Adam with $\beta_1$ set to 0.5 and $\beta_2$ set to 0.99. Generator's learning rate is 0.0002 and discriminator's learning rate is 0.0001. The same learning rate and momentum were used to train LEAD model. We also add the mixed derivative term with $\alpha_d = 0.3$ and $\alpha_g = 0.0$.

The baseline is a DCGAN with the standard non-saturating loss (non-zero sum formulation). In our experiments, we compute the FID based on 50,000 samples generated from our model vs 50,000 real samples.

**Samples**

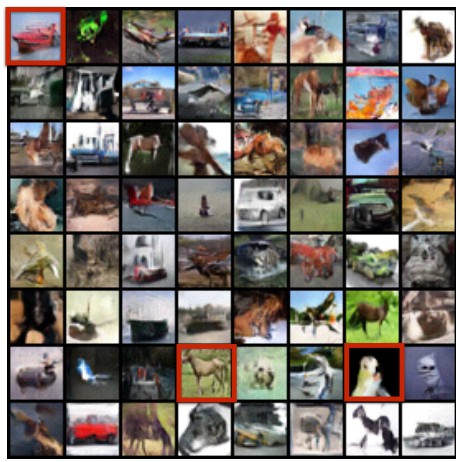 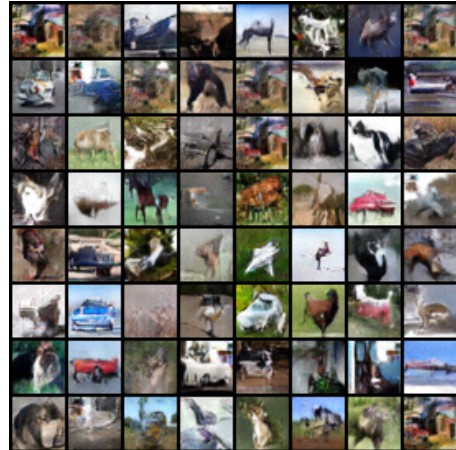

Figure 9: Performance of LEAD on CIFAR-10 image generation task on a DCGAN architecture. **Left**: LEAD achieves FID 19.27. **Right**: Vanilla Adam achieves FID 24.38. LEAD is able to generate better sample qualities from several classes such as ships, horses and birds (red). Best performance is reported after 100 epochs.

### J.3 CIFAR 10 ResNet

**Dataset** The CIFAR10 dataset is available for download at the following link; `https://www.cs.toronto.edu/~kriz/cifar.html`

**Architecture** See Table 6 for a schematic of the architecture used for the CIFAR10 experiments with ResNet.

Table 5: ResNet blocks used for the ResNet architectures (see Table 6).

| **Gen–Block** | **Dis–Block** |
|---|---|
| *Shortcut*: | *Shortcut*: |
| Upsample($\times$2) | downsample |
| *Residual*: | conv. (ker: 1$\times$1, $3_{\ell=1}/128_{\ell\neq1} \to 128$; stride: 1) |
| Batch Normalization | Spectral Normalization |
| ReLU | [AvgPool (ker:2$\times$2, stride:2)], if $\ell \neq 1$ |
| Upsample($\times$2) | *Residual*: |
| conv. (ker: 3$\times$3, 256 $\to$ 256; stride: 1; pad: 1) | [ ReLU ], if $\ell \neq 1$ |
| Batch Normalization | conv. (ker: 3$\times$3, $3_{\ell=1}/128_{\ell\neq1} \to 128$; stride: 1; pad: 1) |
| ReLU | Spectral Normalization |
| conv. (ker: 3$\times$3, 256 $\to$ 256; stride: 1; pad: 1) | ReLU |
| | conv. (ker: 3$\times$3, 128 $\to$ 128; stride: 1; pad: 1) |
| | Spectral Normalization |
| | AvgPool (ker:2$\times$2 ) |

**Other Details** The baseline is a ResNet with non-saturating loss (non-zero sum formulation). Similar to (Miyato et al., 2018), for every time that the generator is updated, the discriminator is updated 5 times. For both the Baseline SNGAN and LEAD-Adam we use a $\beta_1$ of 0.0 and $\beta_2$ of 0.9 for Adam. Baseline SNGAN uses a learning rate of 0.0002 for both the generator and the discriminator. LEAD-Adam also uses a learning rate of 0.0002 for the generator but 0.0001 for the discriminator. LEAD-Adam uses an $\alpha$ of 0.5 and 0.01 for the generator and the discriminator respectively. Furthermore, we evaluate both the baseline and our method on an exponential moving average of the generator's parameters.

Table 6: ResNet architectures used for experiments on CIFAR10.

| Generator | Discriminator |
|---|---|
| *Input: $z \in \mathbb{R}^{64} \sim \mathcal{N}(0, I)$* | *Input: $x \in \mathbb{R}^{3 \times 32 \times 32}$* |
| Linear($64 \rightarrow 4096$) | D–ResBlock |
| G–ResBlock | D–ResBlock |
| G–ResBlock | D–ResBlock |
| G–ResBlock | D–ResBlock |
| Batch Normalization | ReLU |
| ReLU | AvgPool (ker:8×8 ) |
| conv. (ker: 3×3, $256 \rightarrow 3$; stride: 1; pad:1) | Linear($128 \rightarrow 1$) |
| $Tanh(\cdot)$ | Spectral Normalization |

In our experiments, we compute the FID based on 50,000 samples generated from our model vs 50,000 real samples and reported the mean and variance over 5 random runs. We have provided pre-trained models as well as the source code for both LEAD-Adam and Baseline SNGAN in our GitHub repository.

**Samples**

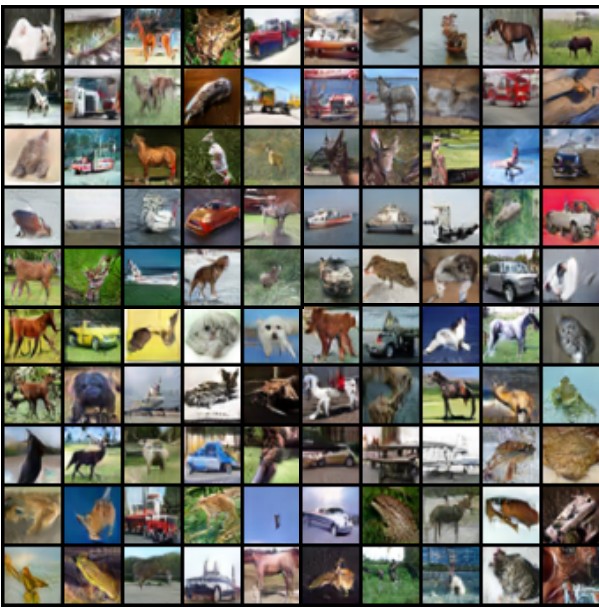

Figure 10: Generated sample of LEAD-Adam on CIFAR-10 after 50k iterations on a ResNet architecture. We achieve an FID score of 10.49 using learning rate $2e - 4$ for the generator and the discriminator, $\alpha$ for the generator is set to 0.01 and for the discriminator is set to 0.5.

## K    Comparison to other methods

In this section we compare our method with several other second order methods in the min-max setting.

The distinction of LEAD from SGA and LookAhead, can be understood by considering the $1^{\text{st}}$-order approximation of $x_{k+1} = x_k - \eta \nabla_x f(x_k, y_k + \eta \Delta y_k)$, where $\Delta y_k = \eta \nabla_y f(x_k + \eta \Delta x, y_k)$.

---

[4]For FtR, we provide the update for the second player given the first player performs gradient descent. Also note that in this table SGA is simplified for the two player zero-sum game. Non-zero sum formulation of SGA such as the one used for GANs require the computation of $\mathbb{J}\mathbf{v}, \mathbb{J}^\top \mathbf{v}$.

Table 7: Comparison of several second-order methods in min-max optimization. Each update rule, corresponding to a particular row, can be constructed by adding cells in that row from Columns 4 to 7 and then multiplying that by the value in Column 1. Furthermore, $\Delta x_{k+1} = x_{k+1} - x_k$, while $\mathcal{C} = \left(I + \eta^2 \nabla^2_{xy} f \nabla^2_{yx} f\right)$. We compare the update rules of the first player[4] for the following methods: Gradient Descent-Ascent (GDA), Least Action Dynamics (LEAD, ours), Symplectic Gradient Adjustment (SGA), Competitive Gradient Descent (CGD), Consensus Optimization (CO), Follow-the-Ridge (FtR) and Learning with Opponent Learning Awareness (LOLA), in a zero-sum game.

| | | Coefficient | Momentum | Gradient | Interaction-xy | Interaction-xx |
|---|---|---|---|---|---|---|
| GDA | $\Delta x_{k+1} =$ | 1 | 0 | $-\eta\nabla_x f$ | $-\eta\nabla_x f$ | 0 |
| **LEAD** | $\mathbf{\Delta x_{k+1}} =$ | **1** | $\beta\mathbf{\Delta x_k}$ | $-\eta\nabla_{\mathbf{x}}\mathbf{f}$ | $-\alpha\nabla^2_{\mathbf{xy}}\mathbf{f}\mathbf{\Delta y_k}$ | 0 |
| SGA[6] | $\Delta x_{k+1} =$ | 1 | 0 | $-\eta\nabla_x f$ | $-\eta\gamma\nabla^2_{xy}f\nabla_y f$ | 0 |
| CGD[53] | $\Delta x_{k+1} =$ | $\mathcal{C}^{-1}$ | 0 | $-\eta\nabla_x f$ | $-\eta^2\nabla^2_{xy}f\nabla_y f$ | 0 |
| CO[39] | $\Delta x_{k+1} =$ | 1 | 0 | $-\eta\nabla_x f$ | $-\eta\gamma\nabla^2_{xy}f\nabla_y f$ | $-\eta\gamma\nabla^2_{xx}f\nabla_x f$ |
| FtR[57] | $\Delta y_{k+1} =$ | 1 | 0 | $\eta_y\nabla_y f$ | $\eta_x\left(\nabla^2_{yy}f\right)^{-1}\nabla^2_{yx}f\nabla_x f$ | 0 |
| LOLA[15] | $\Delta x_{k+1} =$ | 1 | 0 | $-\eta\nabla_x f$ | $-2\eta\alpha\nabla_{xy}f\nabla_y f$ | 0 |

This gives rise to:

$$x_{k+1} = x_k - \eta\nabla_x f\left(x_k, y_k\right) - \eta^2\nabla^2_{xy}f\left(x_k, y_k\right)\Delta y, \tag{67}$$

$$y_{k+1} = y_k + \eta\nabla_y f\left(x_k, y_k\right) + \eta^2\nabla^2_{xy}f\left(x_k, y_k\right)\Delta x, \tag{68}$$

with $\Delta x, \Delta y$ corresponding to each player accounting for its opponent's potential next step. However, SGA and LookAhead additonally *model* their opponent as *naive* learners i.e. $\Delta x = -\nabla_x f(x_k, y_k)$, $\Delta y = \nabla_y f(x_k, y_k)$. On the contrary, our method does away with such specific assumptions, instead modeling the opponent based on its most recent move.

Furthermore, there is a resemblance between LEAD and OGDA that we would like to address. The 1$^{\text{st}}$ order Taylor expansion of the difference in gradients term of OGDA yields the update (for $x$):

$$x_{k+1} = x_k - \eta\nabla_x f - \eta^2\nabla^2_{xy}f\nabla_y f + \eta^2\nabla^2_{xx}f\nabla_x f, \tag{69}$$

which contains an extra 2$^{\text{nd}}$ order term $\nabla^2_{xx}f$ compared to ours. As noted in Schäfer & Anandkumar (2019), the $\nabla^2_{xx}f$ term does not systematically aid in curbing the min-max rotations, rather causing convergence to non-Nash points in some settings. For e.g., let us consider the simple game $f(x, y) = \gamma(x^2 - y^2)$, where $x, y, \gamma$ are all scalars, with the Nash equilibrium of this game located at $(x^* = 0, y^* = 0)$. For a choice of $\gamma \geq 6$, OGDA fails to converge for any learning rate while methods like LEAD, Gradient Descent Ascent (GDA) and CGD (Schäfer & Anandkumar (2019)) that do not contain the $\nabla_{xx}f(\nabla_{yy}f)$ term do exhibit convergence. See Figure 11 and Schäfer & Anandkumar (2019) for more discussion.

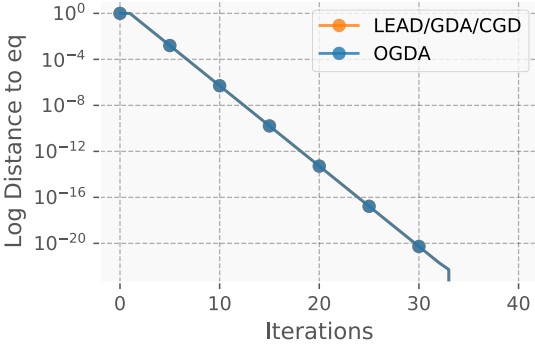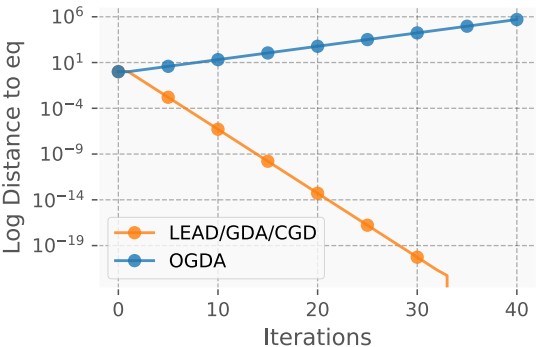

Figure 11: Figure depicting the convergence/divergence of several algorithms on the game of $f(x, y) = \gamma(x^2 - y^2)$ (Nash equilibrium at $x^* = 0, y^* = 0$). **Left**: For $\gamma = 1$, OGDA and LEAD/GDA/CGD (overlaying) are found to converge to the Nash eq. **Right**: For $\gamma = 6$, we find that OGDA fails to converge while LEAD/GDA/CGD (overlaying) converge. We conjecture that the reason behind this observation is the existence of $\nabla^2_{xx}f$ term in the optimization algorithm of OGDA.

