# OpenReview forum: "LEAD: Min-Max Optimization from a Physical Perspective"
_TMLR — Accepted by TMLR_

### Review · Reviewer_8GPk · 2023-03-27

**Summary Of Contributions:**

Inspired by the intuition behind Polyak's momentum methods, which interpret the process of gradient-based optimization as an object (ball) rolling downhill to reach a minimum, the author introduces a similar physics formalism for two-player zero-sum games. They apply "forces" to the object in Polyak's physical intuition, which helps curb rotations and limit speed. Based on this intuition, the author proposes the Least Action Dynamics (LEAD) optimization scheme, which describes these "forces."

The work then extends Lyapunov analysis from momentum methods to LEAD, proving its linear convergence in both continuous and discrete-time settings for quadratic min-max games. Experimentally, the author demonstrates that LEAD is computationally efficient and improves the performance of GANs against multiple baselines across various tasks. The main contributions are as follows:

(1)	LEAD is well-motivated by the physical intuition behind the common momentum method.

(2)	Lyapunov stability analysis guarantees linear convergence speed in both discrete and continuous-time settings for min-max games.

(3)	Experiments effectively support the performance claims made by the author.


**Audience:**

Yes

**Broader Impact Concerns:**

There is no concern to be addressed as far as I know.

**Claims And Evidence:**

Yes

**Requested Changes:**

It is difficult to determine how the discretization parameter affects performance. It would be beneficial if you could display your parameters in the experimental section or provide empirical justification.

As stated immediately before Equation 19, the fixed point $\omega^*$ satisfies $F_{\eta}(\omega^*) = \omega^*$. Does this lead to Equation 19 when $\nabla v(\omega^*)$ (the second term on the RHS of Eq 19) is equal to zero?

The spectrum of $\nabla F_{\eta}(\omega^*)$ is defined in Eq. 20, but the definition of “Sp(off-diag(.))” remains unclear.

The attached blog images in the supplementary, which depict the dynamics in the x-y plane, effectively illustrate the outward rotatory motion and increasing velocity. It would be advantageous to include these images in the main text if possible.


**Strengths And Weaknesses:**

Strengths:

(1)	The motivation rooted in physics is robust and easily comprehensible.

(2)	The proof is overall clear and persuasive.

(3)	Computationally efficient – it does not require the entire Jacobian matrix, but only a quarter block of it. This "partial" Jacobian can be computed using auto-grad tools.

Weaknesses:

The scope of applicability is unclear. This work focuses on two-player quadratic games, specifically requiring an objective function in the form of Eq. 13 in the main text. It is not evident whether the method works for generalized quadratic objective functions, such as objective functions in the form of $X^{T}AX + Y^{T}BY + X^{T}CY$, where A, B, and C are not guaranteed to be positive-definite (in which case the Nash equilibrium may not be zero).

---

> ### Author Response · Authors · 2023-04-19
>
> We would to thank you for your detailed comments and feedback. We are glad that you have found our work "well-motivated". Also, thank you for your positive feedback, "Experiments effectively support the performance claims made by the author", it is greatly appreciated.
>
> Below we clarify a few points:
>
> - "It is difficult to determine how the discretization parameter affects performance." It would be beneficial if you could display your parameters in the experimental section or provide empirical justification.
>
> We have shown in Proposition 1, the connection between the learning rate and the discretization step: ($\eta = \delta^2$). Thus, in all our experimental set-up where we report the learning rate, the discretization step can also be computed given above.
> Furthermore, we have included a new plot in Appendix G, where we provide a grid of experiments to study the effect of discretization on the convergence of the quadratic game explored in section 7.1 (Adversarial vs Cooperative games). We observe that for every level of $\gamma_max$ that changes the dynamics of the game from adversarial to cooperative game, LEAD allows for a wide range of discretization steps. Note that the discretization step and consequently the learning rate can be viewed as a hyper-parameter and consequently require tuning.
>
> - "As stated immediately before Equation 19, the fixed point $\omega^*$ satisfies $F_\eta (\omega^*) = \omega^*$, Does this lead to Equation 19 when $\nabla v(\omega^*)$ is equal to zero?"
>
> We would like to point out that v is the vector field, defined in eq. 14, which is zero at the fixed point (thus $F_\eta(\omega^*) = \omega^*$). However, in equation 19, we are computing  $\nabla v(\omega^*)$ which is the **Jacobian of the vector field** which may not be zero at the fixed point.
>
> - "The spectrum of $\nabla F_\eta(\omega^*)$ is defined in Eq. 20, but the definition of “Sp(off-diag(.))” remains unclear.
>
> Thanks for pointing this out! We have now defined Sp(off-diag(.)) in the notation section.
>
> - "The attached blog images in the supplementary, which depict the dynamics in the x-y plane, effectively illustrate the outward rotatory motion and increasing velocity. It would be advantageous to include these images in the main text if possible."
>
> We are glad that you have studied our blog-post and thank you for the suggestion. As also suggested by reviewer nSLp, we have included the plots in the blog-post in the main text. See Section 3 and Figure 1.
>
> - "This work focuses on two-player quadratic games, specifically requiring an objective function in the form of Eq. 13 in the main text. It is not evident whether the method works for generalized quadratic objective functions, such as objective functions in the form of X^T A X + Y^T B Y + X^T C Y where A, B, and C are not guaranteed to be positive-definite (in which case the Nash equilibrium may not be zero)."
>
> We acknowledge the limitation of our theoretical analysis. We provide theoretical analysis in continuous and discrete time for the quadratic min-max game. Furthermore, we provide experimental analysis for other more general types of games. We would like to note that for the mentioned game of X^T A X + Y^T B Y + X^T C Y if A, B and C are positive semi-definite, the Nash would still be at zero, however if A,B or C are negative definite, the Nash equilibrium may not be zero and our analysis does not support this setting. We would like to re-iterate that our work aims to connect game optimization to a physical framework, shedding light on the dynamics of commonly studied games and GANs. We recognize the limitation of our analysis but believe this work is intersting beyond the theoretical analysis.

---

> > ### Comment · Reviewer_8GPk · 2023-05-08
> > **Thank you for the response. It addressed my questions.**
> >
> > I would like to thank the authors for their response. It addressed my questions.

---

### Review · Reviewer_5Tbx · 2023-03-28

**Summary Of Contributions:**

This paper proposes a novel optimization method to solve min-max game optimization problems, and the proposed method leverages physical properties, e.g., mechanism of forces, to design the proposed optimization methods.

**Audience:**

Yes

**Broader Impact Concerns:**

Author stated a possible concern of broader impact in the end of the main paper. I do not have any concerns on the ethical implications of this work.

**Claims And Evidence:**

Yes

**Requested Changes:**

There are some trivial and non-trivial issues I found in the paper.

$\mathbf{1}.$ It seems as Figure 1 is not referred in the text, and I found its graphic/notation is a bit confusing. It'd be nice to add its context in the text and make its explanation clearer.

$\mathbf{2}.$ In the experiment, authors mention "State-of-the-art methods use architectures that are 30 times or more larger than the architecture that we have chosen to test our method on”. Is it necessary to evaluate the proposed methods on more up-to-date models/instances? If so, it'd be a good practice to add additional experiments into the paper. If not, please explain.

$\mathbf{3}.$ I understand there exists a section discussing the computational cost of proposed algorithm again other existing methods. However, in the experiment, e.g., when comparing Adam vs. LEAD, it would be good to add either cost of computation (e.g., # of gradient) or wall clock time.

Trivial one:

$\mathbf{4}.$ in Section 7.2 on page 10, “See H for a comparison” missed “Appendix”.

**Strengths And Weaknesses:**

This is paper is very well written and organized. The novelty of the proposed method lies on the connection between optimization method with momentum and physical laws. I am not aware of such explicit proposal in the literature.

$\mathbf{1}.$ The technical analysis is thorough where authors considered both discrete and continuous cases, and the convergence analysis is sound and solid.

$\mathbf{2}.$ The novel proposal of connecting momentum with physics in solving min-max game can be considered as a solid contribution, as well as the intuitive design of the algorithm.

$\mathbf{3}.$ The related work is very well stated with detailed background information and lines of research. The motivation and physical connection are very well stated with intuitive illustrations, mainly on Section $3$. Authors also conduct a good comparison with existing algorithms especially on the computational cost.

$\mathbf{4}.$ The experiment is very comprehensive as I see, as well as being supported by comprehensive details, such as the experiment setup. The appendix has extensive details that helps get further understanding of the empirical experiments. I see the proposed toy experiment on the min-max problem in Eq. (31) is very helpful in understanding the practical convergence between LEAD and other existing algorithms. Followed by the toy problem, the experiments on GAN is also presented and designed well to support the claim of the paper.

$\mathbf{5}.$ Author provided extensive details in appendix as well as a blog post of a nice illustration in the supplementary material. I found it very helpful to understand the problem to be solved and the proposed algorithm.

--------------------

Regarding the weakness, I defer it to the next section where I'd like to ask questions and propose a few changes. $\textbf{To highlight my major concern}$, I see the proposed algorithm performs well on top of Adam on a few GAN, but I'd like to understand more of its general applicability during discussion, as it could be potentially the limitation of the proposed method.

---

> ### Author Response · Authors · 2023-04-19
>
> We would like to thank 5Tbx for their encouraging comments. We are glad that you found our paper "a solid contribution", "very well written and organized", "very well stated with detailed background information".
>
> Below we clarify a few points:
> -  "Figure 1 is not referred to in the text, and I found its graphic/notation is a bit confusing. It'd be nice to add its context in the text and make its explanation clearer."
>
> Thank you for the suggestion! We have modified the text to include a description of Figure 1.
> In short, Figure 1 compliments the spectral analysis discussed in section 4.2. According to Proposition 2, an update operator F, is linearly convergent if its spectral radius is smaller than one. Also Theorem 2 states that the LEAD operator has two eigenvalues $\mu_{+}$ and $\mu_{-}$ for each $\lambda \in \text{Sp}\left(\text{off-diag}[\nabla {{v}} ({\omega}^*)]\right)$. Specifically, $\mu_{+}$ can be viewed as a shift of the eigenvalues of GDA while additionally being the leading eigenvalue for small values of $h, \eta, |\alpha|$ and $|\beta|$, the goal in Figure 1 is to show this shift.
>
> Specifically, in Figure 1, the black circle is the unit circle. Thus we would need the eigenvalues to be inside this circle to have convergent dynamics. We plot the eigenvalues of the update operator of Gradient descent ascent in blue and show that they are outside the unit circle. We also plot the eigenvalues of the update operator of LEAD and show that they are inside the unit circle (or linearly convergent) for a wide range of learning rates (moving on the red curves).
>
> - "In the experiment, authors mention "State-of-the-art methods use architectures that are 30 times or more larger than the architecture that we have chosen to test our method on”. Is it necessary to evaluate the proposed methods on more up-to-date models/instances? If so, it'd be a good practice to add additional experiments into the paper. If not, please explain."
>
> Our work aims to connect game optimization to a physical framework, shedding light on the dynamics of commonly studied games and GANs. While previous studies of game dynamics have focused on smaller-scale versions of these games and GANs [1, 2, 3], our approach provides a more comprehensive understanding. We have demonstrated the effectiveness of our method on two GAN architectures (DCGAN and ResNet), but our primary objective is not to surpass state-of-the-art generative models like StyleGAN models (which contain more than 500 million parameters). Therefore, we recognize that scaling our approach to larger architectures requires in-depth experiments, which we hope to explore in future research.
>
> - "I understand there exists a section discussing the computational cost of the proposed algorithm against other existing methods. However, in the experiment, e.g., when comparing Adam vs. LEAD, it would be good to add either cost of computation (e.g., # of gradient) or wall clock time."
>
> Thanks for mentioning this point. We have modified the caption in Figure 4 to provide more context when comparing our method versus vanilla adam.
>
> - "in Section 7.2 on page 10, “See H for a comparison” missed “Appendix”."
>
> Thank you! It is fixed in the updated version.
>
> - "I see the proposed algorithm performs well on top of Adam on a few GAN, but I'd like to understand more of its general applicability during discussion, as it could potentially be the limitation of the proposed method."
>
> We would like to emphasize that our approach, LEAD, has been experimentally validated on different games. In Section 7.1, we evaluated LEAD in a toy setup with various adversarial and cooperative components, while in Section 7.2, we applied it to two different GAN architectures, and in Section H, we used it to optimize the 8-Gaussians. Furthermore, as discussed in Section 9 (Conclusion and Future Direction), we believe that LEAD has potential for use in other types of games. Specifically, since our approach models second-order interactions between players, it may be particularly effective in games with higher-order structure, including any 2-parameter dynamical system, which can be interpreted as a 2-player game. Thus, future research could explore the applicability of LEAD to these types of games.
>
> [1] Panayotis Mertikopoulos, Bruno Lecouat, Houssam Zenati, Chuan-Sheng Foo, Vijay Chandrasekhar, and Georgios Piliouras. Optimistic mirror descent in saddle-point problems: Going the extra (gradient) mile. arXiv preprint arXiv:1807.02629, 2018
>
> [2] Florian Schäfer and Anima Anandkumar. Competitive gradient descent. In Advances in Neural Information Processing Systems, pp. 7623–7633, 2019.
>
> [3] Gauthier Gidel, Hugo Berard, Gaëtan Vignoud, Pascal Vincent, and Simon Lacoste-Julien. A variational inequality perspective on generative adversarial networks. In International Conference on Learning Representations, 2018

---

> > ### Comment · Reviewer_5Tbx · 2023-05-22
> > **Thank authors for the detailed feedback and fix!**
> >
> > I thank the authors for the detailed feedback and I believe they fixed the issue and addressed all of my concerns.

---

### Review · Reviewer_nSLp · 2023-04-09

**Summary Of Contributions:**

This paper proposes a new optimization algorithm called LEAD for min-max optimization. The algorithm is developed by observing gradient descent dynamics, drawing analogies with a physical system, and modifying it using momentum and a coupling term. The convergence of LEAD is analyzed on a quadratic min-max optimization problem. It converges exponentially and is asymptotically quicker than an extra-gradient method in some regimes. The paper also provides a detailed empirical study of the computational cost of implementing LEAD v/s other methods, showing LEAD can be implemented with two gradient computations at each step. Finally, the paper compares the optimization of LEAD v/s previous methods on a family of synthetic and well-known non-convex tasks, showing either improved or comparable performance.

**Overall, I like the writing and the flow of the paper and recommend accepting it.**

**Audience:**

Yes

**Broader Impact Concerns:**

None. I appreciate the authors adding a broader impact section.

**Claims And Evidence:**

Yes

**Requested Changes:**

1. Add an intuitive example discussing the effect of the magnetic forces. This is not a big issue but might help provide further physical intuition.

2. Add theoretical comparison to other methods. A table earlier in the paper discussing the convergence rates for all the methods would be most desirable. This should be discussed for all the algorithms used in Figure three. This is an important addition.

3. Discuss any difficulties in extending the results to the general convex-concave games.



**Strengths And Weaknesses:**

1. The paper is well written. It starts with a discussion of the optimization mechanics, providing good intuition for how the optimization method was developed by introducing two different kinds of "forces" and showing their effect on the final update equations of the algorithm. Making an analogy to Newton's second law and using (for the most part) familiar physical terminology was helpful. I didn't fully understand why magnetic force does what the authors describe, and it might be good to add an illustrative physical (or abstract) example in the paper.

2. The rotational dynamics hindering the convergence of gradient descent ascent is a well-talked-about phenomenon, and several algorithms have been proposed to avoid it. I found the discussion of the existing algorithms a bit lacking. In particular, while many works were cited, the precise convergence rates were not discussed. This makes it difficult to put the theoretical results in this paper in perspective.

3. The authors don't discuss if their results can be extended to the general quadratic setting and, more broadly, to convex-concave games. I think this is a major limitation of the paper and requires discussion.

---

> ### Author Response · Authors · 2023-04-21
>
> We would like to thank nSLp for their encouraging comments. We are glad that you liked the writing and the flow of the paper. Below we clarify a few points:
> - "I didn't fully understand why magnetic force does what the authors describe, and it might be good to add an illustrative physical (or abstract) example in the paper." and "Add an intuitive example discussing the effect of the magnetic forces"
>
> In our work we used the magnetic force to curb the rotational dynamics induced by the vortex force. Our inspiration came from the observation that when a particle with charge q moves within a magnetic field, the magnetic force is applied to it in a direction perpendicular to the current direction of the particle. This magnetic force induces a rotational dynamic in the particle, the direction of which can be altered by the sign of the charge (q) of the particle. We selected the sign of the charge such that it would reduce the overall rotation of the particle. As also suggested by reviewer 8GPk, we have included the plots in the blog-post in the main text. See Section 3.
>
> - "Add theoretical comparison to other methods. A table earlier in the paper discussing the convergence rates for all the methods would be most desirable.This should be discussed for all the algorithms used in Figure three. This is an important addition. "
>
> Thank you for the suggestion! We have included Table 1 in Section 3 that compares LEAD with several existing methods for the quadratic min-max game. Note that the rates for the quadratic min-max game do not exist in the literature for many of the methods mentioned in our paper. Please see Table 1 for more discussion.
>
> We would like to also point out that Figure 3 effectively shows the convergence rates for the game in Eq. 29 as we are plotting the spectral radius. According to Proposition 2, the spectral radius determines the rate of convergence (smaller spectral radius corresponds to faster convergence). For each method in Figure 3, we analytically derive a formula for the spectral radius in terms of the learning rate and then numerically solve the equation and find the learning rate that leads to the smallest spectral radius. We then plot the smallest spectral radius for each method while changing the game's parameter \gamma_max. We observe that LEAD is a robust optimizer across different types of games.
>
> - "Discuss any difficulties in extending the results to the general convex-concave games."
>
> Our existent analysis and the subsequent promising experimental results there upon, makes us believe that analysis can be extended to the more general convex-concave setting. Nevertheless, such an attempt is dependent on finding an appropriate Lyapunov function, which till now has been challenging, and is the subject of future work. In general, Lyapunov analysis can be a challenging task because it requires finding a Lyapunov function that satisfies certain conditions, such as being positive definite and having a negative definite derivative along the trajectories of the system. The analysis for our particular setup becomes even more challenging with the game dynamics where previously known Lyapunov functions or their modified versions do not directly apply. We have modified the text in Section 9: Conclusion and Future Direction, to include this point.

---

### Author Response · Authors · 2023-04-21
**To all the reviewers**

We are glad that all the reviews have found our work relevant to the TMLR community, well-motivated and well-written. Below is a summary of the changes that are marked in red in the updated pdf document:
- Including new plots from the blog-post in the main text in Section 3.
- Including Table 1, in section 3 that compares the convergence rate of LEAD vs several existing methods for the quadratic min-max game.
- Including a new figure in Appendix G that studies the effect of discretization step-size on the performance of LEAD.
- Updating the conclusion and Future Direction section to include: 1) difficulties of extending the results to general convex-concave games. 2) general applicability of LEAD in other games.
- Including a description of Figure 1 in Section 4.2.
- There was also a minor error in our proof of Th.3. which is fixed now. Note that this change does not affect our results or findings. We have marked the changes in the pdf in red.
- Other minor points mentioned by the reviews.

Thanks again for your valuable comments and constructive feedbacks!

---

### Decision · Action_Editors · 2023-05-23

**Recommendation:** Accept with minor revision

**Comment:**

This paper proposes a physics-inspired formalism for two-player zero-sum games and an optimization scheme, called LEAD. The work applies Lyapunov analysis to LEAD, proving its linear convergence in both continuous and discrete-time settings for quadratic min-max games. The connection to physics is well grounded and the theoretical contributions are solid. Besides that, the authors demonstrate experimentally that LEAD is computationally efficient and improves the performance of GANs. The paper contains a lot of details that show evidence of the empirical performance.

The reviewers found the contributions of this work are solid. Moreover, the paper is very well written. All reviewers unanimously voted for acceptance.

Therefore, the paper is ready for acceptance almost in its current form. Please make sure to remove the red font from the text. I also found a relevant reference that I think should be included in the final version of the paper: https://arxiv.org/abs/2105.13922

**Audience:**

Yes: the paper is of interest for researchers working on min-max optimization and adversarial training, and potentially to those with a physics background who are interested in exploring connections between physics and algorithmic contributions in machine learning.

**Claims And Evidence:**

The paper proposes a physics-inspired formalism for two-player zero-sum games and an optimization scheme, called LEAD. The authors  demonstrate experimentally that LEAD is computationally efficient and improves the performance of GANs. The paper contains a lot of detail that show evidence of the empirical performance. Moreover, the theoretical contributions and proofs are clear and persuasive.